# Structure and physiological function of the human KCNQ1 channel voltage sensor intermediate state

Keenan C Taylor[1,2†], Po Wei Kang[3†], Panpan Hou[3†], Nien-Du Yang[3], Georg Kuenze[2,4], Jarrod A Smith[1,2], Jingyi Shi[3], Hui Huang[1,2], Kelli McFarland White[3], Dungeng Peng[1,2,5], Alfred L George[6], Jens Meiler[2,4,7], Robert L McFeeters[8], Jianmin Cui[3*], Charles R Sanders[1,2,9*]

[1]Department of Biochemistry, Vanderbilt University, Nashville, United States; [2]Center for Structural Biology, Vanderbilt University, Nashville, United States; [3]Department of Biomedical Engineering, Center for the Investigation of Membrane Excitability Disorders, and Cardiac Bioelectricity, and Arrhythmia Center, Washington University in St. Louis, St. Louis, United States; [4]Departments of Chemistry and Pharmacology, Vanderbilt University, Nashville, United States; [5]Department of Medicine, Division of Clinical Pharmacology, Vanderbilt University Medical Center, Nashville, United States; [6]Department of Pharmacology, Northwestern University Feinberg School of Medicine, Chicago, United States; [7]Department of Bioinformatics, Vanderbilt University Medical Center, Nashville, United States; [8]Department of Chemistry, University of Alabama in Huntsville, Huntsville, United States; [9]Department of Medicine, Vanderbilt University Medical Center, Nashville, United States

**\*For correspondence:**
jcui@wustl.edu (JC);
chuck.sanders@vanderbilt.edu (CRS)

†These authors contributed equally to this work

**Abstract** Voltage-gated ion channels feature voltage sensor domains (VSDs) that exist in three distinct conformations during activation: resting, intermediate, and activated. Experimental determination of the structure of a potassium channel VSD in the intermediate state has previously proven elusive. Here, we report and validate the experimental three-dimensional structure of the human KCNQ1 voltage-gated potassium channel VSD in the intermediate state. We also used mutagenesis and electrophysiology in *Xenopus laevis*oocytes to functionally map the determinants of S4 helix motion during voltage-dependent transition from the intermediate to the activated state. Finally, the physiological relevance of the intermediate state KCNQ1 conductance is demonstrated using voltage-clamp fluorometry. This work illuminates the structure of the VSD intermediate state and demonstrates that intermediate state conductivity contributes to the unusual versatility of KCNQ1, which can function either as the slow delayed rectifier current ($I_{Ks}$) of the cardiac action potential or as a constitutively active epithelial leak current.

## Introduction

Voltage-gated potassium ($K_V$) channels are critical for electrical signaling in excitable cells where they drive action potential termination. In $K_V$ channels, the voltage sensor domains (VSDs) undergo specific conformational changes during membrane depolarization to activate channel opening. Previous studies revealed that $K_V$ VSDs activate sequentially from the initial resting state to an experimentally resolvable intermediate state and then to the activated state (*Sigworth, 1994*; *Bezanilla et al., 1994*; *Silva et al., 2009*; *Zagotta et al., 1994*; *Baker et al., 1998*; *Jensen et al., 2012*; *Lacroix et al., 2012*; *Silverman et al., 2003*; *Barro-Soria et al., 2014*; *Wu et al., 2010*;

*Zaydman et al., 2014*; *Hou et al., 2017*; *Osteen et al., 2012*; *Swartz, 2008*; *Li et al., 2014a*; *Roux, 2006*; *Sigg et al., 1994*; *Vargas et al., 2012*; *Hou et al., 2019*; *Hou et al., 2020*; ). The movement associated with VSD activation then induces the channel pore domain to open and conduct ionic current. Accordingly, the structural basis underlying VSD conformational change during activation constitutes a fundamental aspect of $K_V$ channel voltage-dependent gating. Despite the importance of VSDs in voltage-dependent gating, an experimental structure for the intermediate state of a $K_V$ VSD has not been reported. Here, we present and functionally validate the three-dimensional structure of the human voltage-gated potassium channel KCNQ1 ($K_V$7.1) VSD in the intermediate state.

Although numerous high-resolution structures of voltage-gated ion channels VSD are available (*Li et al., 2014a*; *Long et al., 2005a*; *Long et al., 2005b*; *Sun and MacKinnon, 2017*; *Li et al., 2014b*; *Kintzer and Stroud, 2016*; *Shen et al., 2019*; *Clairfeuille et al., 2019*; *Wisedchaisri et al., 2019*), all experimental VSD structures were determined at 0 mV membrane potential due to the inability to control the membrane potential in the model membrane media used for structural studies. Because 0 mV represents a physiologically depolarized potential for voltage-gated ion channels, nearly all structures of VSDs are thought to represent the depolarized or 'up' conformation. This technical challenge has historically complicated structure determination for VSDs in intermediate and resting state conformations. Nevertheless, prior studies have applied varied strategies to characterize VSD structures of voltage-gated channels in alternate conformations. These studies have employed a variety of approaches, including exploiting metal affinity cross-linking to resolve the HCN channel VSD in the hyperpolarized conformation (*Lee and MacKinnon, 2017*), applying site-directed mutagenesis and cysteine crosslinking to bias a $Na_V$ VSD into the resting conformation (*Wisedchaisri et al., 2019*), utilizing a VSD-binding toxin to trap a $Na_V$ VSD in the deactivated state (*Clairfeuille et al., 2019*), and employing $Ca^{2+}$ to bias a TPC1 channel $Ca^{2+}$-sensitive VSD into resting and activated states (*Kintzer and Stroud, 2016*; *Kintzer et al., 2018*). Despite extensive structural studies, an experimental structure for the intermediate state of the KCNQ1 channel VSD has proven elusive. The lack of high-resolution KCNQ1 VSD structures in kinetically significant conformations along the activation pathway represents a major gap in our knowledge of the structural basis of KCNQ1 VSD activation.

A second challenge in structure-function studies of $K_V$ VSDs, such as the KCNQ1 VSD, in non-activated conformations involves functional validation. The most common functional technique to validate $K_V$ channel structures involves measuring ionic currents by voltage or patch clamp experiments. In most $K_V$ channels, it is thought that the pore domain opens to conduct current only upon VSD transition into the fully activated state. This implies that traditional ionic current measurements are blind to VSD occupation of the resting state or the intermediate state, as both VSD states are thought not to induce pore conduction. Following this line of logic, even if high-resolution VSD structures in the intermediate state were to be determined, the lack of straightforward functional electrophysiology tests to discriminate between VSD conformations of non-conducting channel states (e.g. resting state vs. intermediate state) presents a challenge for functional validation. In this regard, it is significant that the VSD of the KCNQ1 $K_V$ channel is thought to populate an intermediate state that promotes a conductive state of the pore domain (*Zaydman et al., 2014*; *Hou et al., 2017*; *Hou et al., 2019*; *Hou et al., 2020*), providing a pathway to functional validation of a VSD structure proposed to represent the intermediate state.

KCNQ1 is a $K_V$ channel that plays multiple physiological roles. When paired with the KCNE1 accessory protein, KCNQ1 provides the delayed-rectifier $I_{Ks}$ current of the cardiac action potential (*Abbott, 2016*; *Barhanin et al., 1996*; *Sanguinetti et al., 1996*; *Hedley et al., 2009*; *Tobelaim et al., 2017*; *Liin et al., 2015*). Loss of function or aberrant gain of function caused by heritable mutations in KCNQ1 causes several different arrhythmias, which include long QT syndrome (LQTS) (*Wu and Sanguinetti, 2016*; *Campuzano et al., 2019*; *Wu et al., 2016*). Alternatively, when paired with accessory protein KCNE3, KCNQ1 plays an important role as a leak channel to help maintain ion homeostasis in epithelial cells (*Abbott, 2016*; *Julio-Kalajzić et al., 2018*; *Schroeder et al., 2000*). KCNQ1 adopts the canonical structural organization of the $K_V$ superfamily in which the central homotetrameric pore domain is flanked by four VSDs, each with four transmembrane helical segments (S1-S4). Each KCNQ1 VSD exhibits sequential activation (*Silva et al., 2009*; *Barro-Soria et al., 2014*; *Wu et al., 2010*; *Zaydman et al., 2014*; *Hou et al., 2020*; *Hou et al., 2017*), similar to other $K_V$ channels such as the *Drosophila* Shaker channel (*Bezanilla et al., 1994*;

*Baker et al., 1998*; *Jensen et al., 2012*; *Lacroix et al., 2012*). However, while both Shaker and KCNQ1 conduct current when their VSDs adopt the activated conformation, KCNQ1 is distinctive in that it can also conduct current when its VSDs occupy the intermediate conformation (*Zaydman et al., 2014*; *Hou et al., 2017*; *Osteen et al., 2012*; *Hou et al., 2019*; *Hou et al., 2020*).

The intermediate conductance of KCNQ1 channels offers an opportunity to overcome the challenge to conventional electrophysiology of discriminating between KCNQ1 VSD in the resting state vs. the intermediate state. Moreover, we have previously shown that the KCNQ1 intermediate and activated conductances feature distinct auxiliary subunit regulation and pharmacology (*Zaydman et al., 2014*; *Hou et al., 2017*; *Hou et al., 2019*; *Hou et al., 2020*). KCNQ1 thus presents an ideal platform for VSD structure-function studies, as traditional electrophysiology techniques can readily distinguish between the resting, intermediate, and activated VSD states. In this study, we determine the structure of the human KCNQ1 VSD and then take advantage of the distinct KCNQ1 intermediate and activated conductances to provide functional evidence that supports this VSD structure as representing the intermediate state rather than the activated or resting states. The cryo-EM structure of the *Xenopus* KCNQ1 determined in dodecylmaltoside (DDM) micelles by the MacKinnon lab appears to represent a channel with a closed pore and flanking VSD domains that populate the fully activated state (*Sun and MacKinnon, 2017*). The MacKinnon lab also determined the structure of human KCNQ1 in complex with KCNE3 (*Sun and MacKinnon, 2020*). The structures of the *Xenopus* and human KCNQ1 VSDs determined by the MacKinnon lab are similar. Whether the VSD in the cryo-EM structures represents the fully activated state is also experimentally addressed in this paper. Lastly, we provide evidence to demonstrate that the conductive intermediate state of the KCNQ1 channel is physiologically relevant and contributes to the channel's functional versatility.

## Results

### NMR structure of the KCNQ1 voltage sensor domain

It has long been known that voltage sensor domains fold autonomously, as reflected by the fact that voltage-gated proton channels are single domain monomeric VSDs (*DeCoursey et al., 2016*; *Ramsey et al., 2006*; *Sasaki et al., 2006*) and also by studies showing that VSDs excised from $K_V$ channels or other voltage-regulated proteins fold independently and yield experimental 3D structures that are consistent with their conformations in the context of intact channels (*Li et al., 2014b*; *Jiang et al., 2003*). Indeed, solution nuclear magnetic resonance (NMR) methods have previously been used to determine the activated state structure of the VSD of the KvAP channel from a hyperthermophilic microorganism (*Shenkarev et al., 2010*; *Butterwick and MacKinnon, 2010*). The NMR-determined structure of the human voltage-gated proton channel $H_V1$ was also recently reported (*Bayrhuber et al., 2019*).

Structural studies of the isolated human KCNQ1 VSD spanning from the S0 segment preceding the S1-S4 transmembrane domain through the middle of the S4-S5 link were undertaken using solution NMR spectroscopy of the protein under conditions where it is solubilized in detergent micelles composed of a lipid-like detergent. Screening of suitable model membrane conditions for solution NMR of the isolated human KCNQ1 VSD was previously described and led to the conclusion that, among the various model membrane conditions tested, micelles formed by lyso-myristoylphosphatidylglycerol (LMPG) or lyso-palmitoylphosphatidylglycerol (LPPG) yielded NMR spectra of superior quality (*Peng et al., 2014*). A similar result was recently reported for preliminary NMR studies of the isolated Shaker channel VSD (*Chen et al., 2019*). The lysophospholipids are among the most phospholipid-like detergents available and are known to be generally mild and non-denaturing (*Koehler et al., 2010*; *Krueger-Koplin et al., 2004*). We conducted the studies of this work in LMPG rather than LPPG micelles (see NMR spectra in *Figure 1*) because a recent study indicated that the wild type KCNQ1 VSD adopts a stable fold in this medium (*Huang et al., 2018*). This was further supported in the present work by the fact that paramagnetic relaxation enhancements (PREs) of spin-labeled VSD samples revealed a transmembrane topology consistent with the voltage sensor fold (*Figure 1—figure supplement 1*). We therefore proceeded with structural studies of the human KCNQ1 VSD in LMPG micelle conditions.

The backbone amide $^1H$, $^{13}C$, and $^{15}N$ resonances and also the side chain methyl peaks of KCNQ1 were assigned using 3D NMR methods (see *Figure 1* and Materials and methods). We then

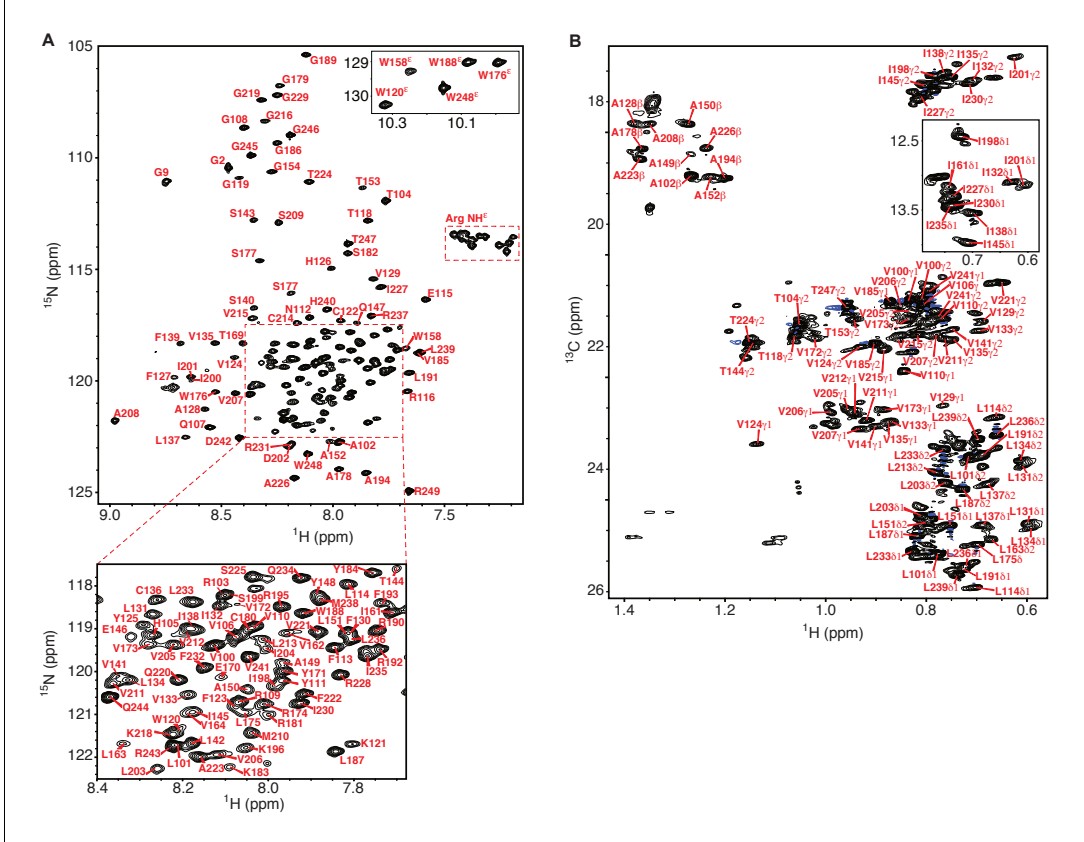

**Figure 1.** NMR spectra of the human KCNQ1 voltage-sensor domain. (A) $^1$H-$^{15}$N TROSY-HSQC spectrum recorded at 900 MHz of $^2$H,$^{13}$C,$^{15}$N-Q1-VSD in LMPG micelles. Backbone amide peaks for 140 out of 147 non-proline residues (95%) have been assigned. Only a single set of peaks is observed. (B) $^1$H-$^{13}$C HSQC methyl optimized spectrum recorded at 900 MHz of $^{13}$C,$^{15}$N-Q1-VSD in perdeuterated LMPG micelles. Methyl groups for 58 out of 68 (85%) residues were assigned (Ala 10 of10, Thr 6 of 8, Ile 9 of 11, Val 18 of 22, Leu 15 of 17). In addition to the presence of a very limited number of unassigned peaks from the VSD in this spectrum other unassigned peaks likely derive from natural abundance $^{13}$C in residually protonated LMPG and also from the fully protonated buffer components TCEP and MES. The chemical shift assignments illustrated for panels A and B shift have been deposited in BioMagResBank (BMRB ID 30517).

The online version of this article includes the following source data and figure supplement(s) for figure 1:

**Figure supplement 1.** Long range PREs are consistent with the expected KCNQ1 VSD topology in LMPG micelles.

**Figure supplement 1—source data 1.** Excel file with numerical data used for *Figure 1—figure supplement 1*.

**Figure supplement 2.** TALOS-N secondary structure analysis of backbone chemical shifts (top panel) and (bottom panels) deviations of the (*Osteen et al., 2012*) Cα and carbonyl $^{13}$C' chemical shifts from random coil values.

**Figure supplement 2—source data 1.** Excel file with numerical data used for *Figure 1—figure supplement 2*.

**Figure supplement 3.** Examples of NOE measurements.

collected a series of distinct classes of NMR restraints as summarized in *Table 1*: backbone torsion angles based on chemical shifts (*Figure 1—figure supplement 2*), short- and long-range $^1$H-$^1$H NOE-derived distances (*Figure 1—figure supplement 3*), long-range distances from PREs, and backbone $^1$H-$^{15}$N residual dipolar couplings (RDCs). PREs involve use of single site spin-labeling to introduce spectroscopic beacons into the VSD that lead to distance-dependent peak broadening. Care was taken to verify that single cysteine mutations and subsequent spin-labeling did not disrupt the protein structure. Indeed, for several sites mutation and/or spin labeling was found to be disruptive of structure, in which cases PRE data was not acquired. While PRE-determined NMR structures have been shown to be robustly reliable (*Liang et al., 2006*; *Gottstein et al., 2012*; *Battiste and Wagner, 2000*; *Ganguly et al., 2011*), long-range NOEs were also incorporated into structure calculations to improve the precision and accuracy of the ensemble. The chemical shift, PRE, and NOE-determined ensemble was refined against measured $^1$H-$^{15}$N backbone RDCs, confirming the

**Table 1.** KCNQ1 VSD NMR structure statistics.

| Structure restraints | XPLOR-NIH* | PDB 6MIE† |
|---|---|---|
| Total NOE | 958 | 958 |
| Inter-residue: | | |
| Sequential ( $\lvert i - j \rvert = 1$ ) | 559 | 559 |
| Medium-range ( $1 < \lvert i - j \rvert < 5$ ) | 366 | 366 |
| Long-range ( $\lvert i - j \rvert \geq 5$ ) | 33 | 33 |
| Hydrogen bonds | 55 | 55 |
| Paramagnetic relaxation enhancement | 403 | 403 |
| Dihedral angle | | |
| $\phi$ | 97 | 97 |
| $\varphi$ | 97 | 97 |
| Residual dipolar couplings ($D_{HN}$) | 54 | 54 |
| | | |
| **Structure statistics** | | |
| Ensemble r.m.s.d. (residues 120–152, 160–239) | | |
| Backbone heavy atoms (Å) | 1.41 | 0.96 |
| All heavy atoms (Å) | 2.33 | 1.72 |
| Transmembrane r.m.s.d. (residues 120–142, 160–179, 198–215, 219–239) | | |
| Backbone heavy atoms (Å) | 0.87 | 0.97 |
| r.m.s.d. from experimental restraints | | |
| Distances (Å) | 0.068 ± 0.005 | 0.150 ± 0.019 |
| Dihedral angles (°) | 1.0 ± 0.2 | 10.3 ± 4.2 |
| Residual dipolar coupling (Hz) | 0.92 ± 0.21 | 3.1 ± 0.6 |
| r.m.s.d. from idealized geometry | | |
| Bond lengths (Å) | 0.003 ± 0.001 | 0.005 ± 0.001 |
| Bond angles (°) | 0.44 ± 0.01 | 1.71 ± 0.01 |
| Ramachandran plot (residues 101–152, 160-239)‡ | | |
| Most favorable (%) | 89.3 ± 2.0 | 89.8 ± 2.8 |
| Additionally allowed (%) | 10.0 ± 2.2 | 8.8 ± 2.3 |
| Generously allowed (%) | 0.4 ± 0.8 | 0.7 ± 0.6 |
| Disallowed (%) | 0.3 ± 0.8 | 0.6 ± 0.7 |

*'XPLOR-NIH' describes the statistics for the XPLOR-NIH structure ensemble generated using experimental restraints, prior to the rMD phase of the calculations.
†'MD' describes the statistics for the structure ensemble (PDB ID: 6MIE) (see Materials and methods).
‡Procheck NMR.

structure with an independent data set. Care also was taken to ensure that no subset of the NOE data had an unduly influential impact on the final ensemble of structures (see Materials and methods).

The ensemble of KCNQ1 VSD structures determined by the NMR data and the XPLOR-NIH program (*Schwieters et al., 2006*) is illustrated in *Figure 2—figure supplement 1A*, with structural statistics in *Table 1*. Because structural studies of membrane proteins in micelles sometimes are complicated by micelle-based distortion of native structure (*Jensen et al., 2012*; *Paramonov et al., 2017*; *Zhou and Cross, 2013*), we took extra steps to account for and correct any such distortions. Specifically, 10 members of the NMR ensemble were selected (based on the root mean squared deviation—r.m.s.d.—to the mean coordinates) for NMR data-restrained molecular dynamics (MD) in a hydrated dimyristoylphosphatidylcholine (DMPC) bilayer. After 100 nsec of restrained MD, the

restraints were turned off and the MD trajectories were allowed to continue for another 190–200 ns to see if the NMR-defined structure would 'hold'. Analysis of an ensemble of 10 centroid structures generated from the final 100 ns of the lowest energy trajectory revealed that this ensemble continued to satisfy the NMR data (*Table 1*). This final VSD structural ensemble is illustrated in *Figure 2A* (PDB ID: 6MIE). *Figure 2—figure supplement 1* shows that the combined restrained/unrestrained molecular dynamics phase of structural refinement resulted in only modest changes relative to the starting XPLOR-NIH NMR conformational ensemble. We emphasize that PDB 6MIE continues to satisfy the NMR restraints (*Table 1*). In this final ensemble, the NMR data defines the protein fold and some side chain conformations. However, the side chain conformations for the key residues highlighted in *Figure 3A* were not directly restrained by any of the experimental data, but were determined by the force fields operative in the XPLOR-NIH simulated annealing protocol and in subsequent MD trajectories.

As will be described later in this paper, functional studies indicate that the VSD structure determined herein (*Figures 2–4*) represents the *intermediate* state conformation along the VSD activation pathway. The preparation of a sample in which the isolated WT VSD occupies the previously structurally uncharacterized intermediate state appears to be the fortuitous consequence of performing studies in LMPG micelles, which stabilizes this otherwise difficult-to-access state, enabling it to be subjected to structural characterization.

The NMR-determined human KCNQ1 VSD conformation features a short surface amphipathic N-terminal helix (S0) and four transmembrane helices (S1-S4), followed by part of the S4-S5 linker, the latter of which was disordered (*Figure 2A*). Comparison of this structure to that of the *Xenopus* KCNQ1 VSD determined by cryo-EM in β-dodecyl-D-maltopyranoside (DDM) micelles (*Figures 2B*, *3* and *4*; *Sun and MacKinnon, 2017*) reveals important differences.

Positively charged amino acids are located along the transmembrane S4 helix of potassium channel VSDs and some of these charges, commonly known as 'gating charges', confer voltage-sensitivity to channel functions. We will refer to the gating charges as R1 through R6, numbered from the N-terminal end to the C-terminal end of the S4 segment (*Figure 3D*). During membrane depolarization, the S4 helix moves from its resting state outward toward the extracellular space (*Glauner et al., 1999*). During this movement, the gating charges successively pair with conserved acidic residues within the VSD (*Papazian et al., 1995*), including residues of the 'charge transfer center', which additionally contains an aromatic residue acting as a 'hydrophobic plug" (*Tao et al., 2010*; *Lacroix and Bezanilla, 2011*). Critical residues that coordinate gating charge movement include the acidic residue E1 (E160 in human KCNQ1) and the charge transfer center residues E2 (E170), D202 and the aromatic plug residue F0 (F167, *Figure 3*). Pairwise electrostatic interactions between the positive gating charges in S4 and the negatively charged S2/S3 residues help solvate the positive S4 residues in the hydrophobic membrane interior to stabilize the VSD. In KCNQ1, electrophysiological and modeling studies suggested that the activated state of the VSD involves pairing of E1 with gating charge site R4 (R237) (*Silva et al., 2009*; *Wu et al., 2010*; *Zaydman et al., 2014*; *Restier et al., 2008*). This pairing was observed in the cryo-EM structure of the KCNQ1 VSD (*Figures 3B* and *4B*; *Sun and MacKinnon, 2017*), suggesting that the VSD seen in that structure reflects the activated state. By inference, the activated state is likely stabilized by additional interactions of the charge transfer residues E2 and F0 on S2 and D202 on S3 with residue H5 (H240) (*Sun and MacKinnon, 2017*). On the other hand, we observed a different arrangement of S4 charge pairings with S2 residues in the NMR structure of the human KCNQ1 VSD. These differences were the consequence of S4 being translated by ~5.4 Å along the bilayer normal toward the extracellular side during the transition from the NMR structure to the cryo-EM structure (*Figure 3A–C*). In the NMR structure, R2 (R231) pairs with E1 (*Figures 3A* and *4A*), which is postulated to be a crucial stabilizing interaction for the intermediate VSD state based on previous electrophysiological results (*Silva et al., 2009*; *Wu et al., 2010*; *Zaydman et al., 2014*). This strongly suggests that the NMR structure represents the intermediate VSD state. Additional observed interactions that likely contribute to intermediate state stabilization include interaction of R4 with charge transfer residues E2 and D202, as well as interaction of Q3 (Q234, corresponding to R3 in most other voltage-gated channels) with F0 (*Figures 3A,D* and *4A*).

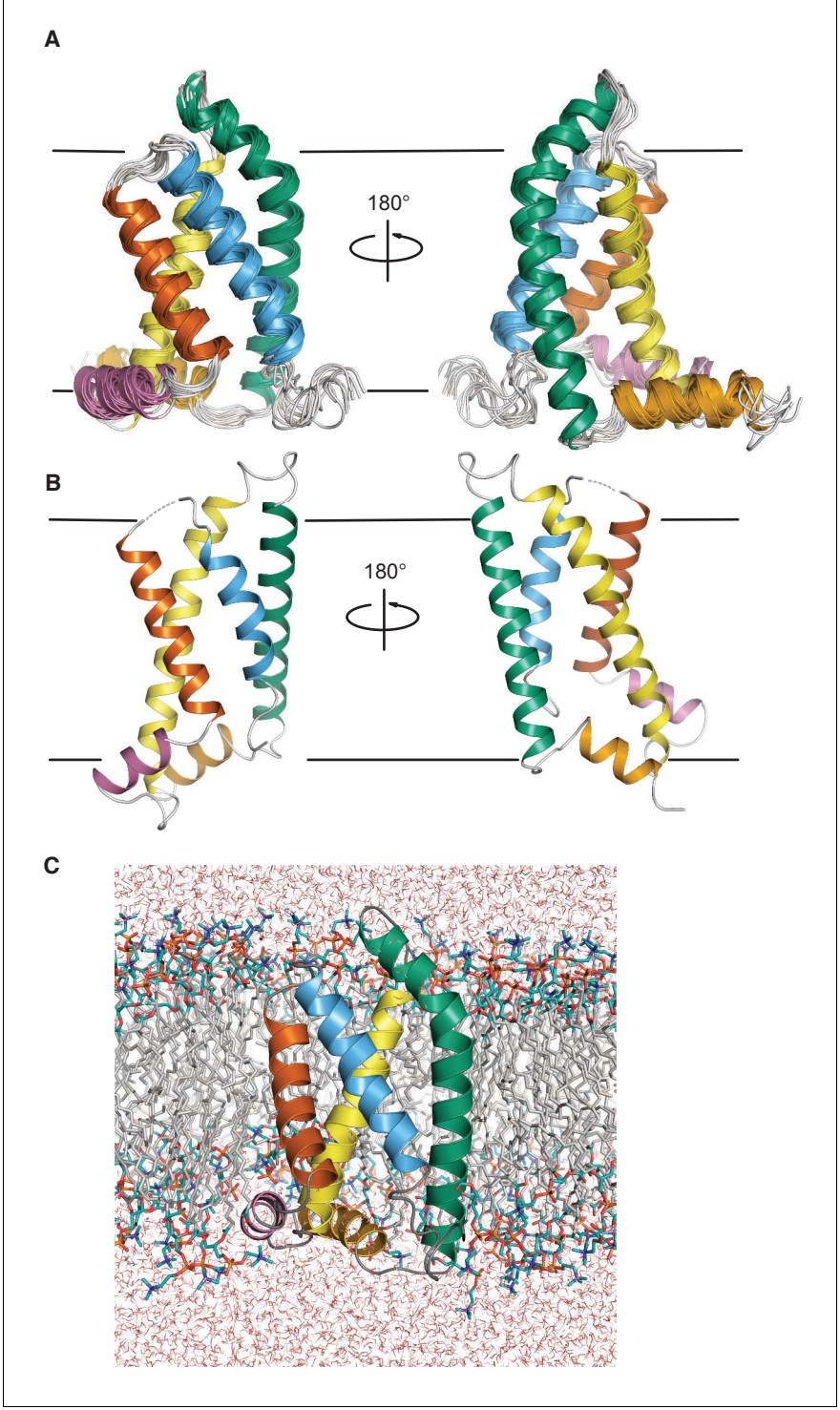

**Figure 2.** Structure of the KCNQ1 VSD. (**A**) The human KCNQ1 VSD NMR-determined ensemble after molecular dynamics refinement in a hydrated DMPC bilayer (PDB ID 6MIE, see statistics in *Table 1* and also *Figure 2—figure supplement 1*) (**B**) Cryo-EM structure of the *Xenopus* KCNQ1 VSD (PDB ID 5VMS) (*Sun and MacKinnon, 2017*). (**C**) Representative low energy NMR structure from PDB 6MIE in a hydrated DMPC bilayer. In panels A-C, the transmembrane helices S1, S2, S3, and S4, are colored bluish green, yellow, vermillion, and sky blue respectively. The S2-S3 linker and S0 helices are colored reddish purple and orange. The approximate position of the membrane-water interfaces is indicated by a pair of black lines in panels A and B.

The online version of this article includes the following source data and figure supplement(s) for figure 2:

*Figure 2 continued on next page*

*Figure 2 continued*

**Figure supplement 1.** KCNQ1-VSD XplorNIH-determined structural ensemble, before and after molecular dynamics refinement.

**Figure supplement 1—source data 1.** Text file with coordinates (PDB format) of the XPLOR-NIH structure ensemble for the KCNQ1-VSD prior to molecular dynamics.

**Figure supplement 1—source data 2.** Excel file with numerical data used for *Figure 2—figure supplement 1C*, top panel.

**Figure supplement 1—source data 3.** Excel file with numerical data used for *Figure 2—figure supplement 1C*, bottom panel.

**Figure supplement 1—source data 4.** Excel file with numerical data used for *Figure 2—figure supplement 1D*, top panel.

## Functional validation of distinct KCNQ1 voltage sensor domain structures

To validate that the NMR structure of the VSD faithfully represents the intermediate state and that the VSD seen in the cryo-EM structure represents the activated state, we tested whether the paired-residue interactions revealed by these two structures can be demonstrated functionally. To this end, we used a double charge reversal mutagenesis strategy (*Figure 5A–C*). Mutation of gating charges in S4 to a negatively charged residue leads to strong electrostatic repulsion between the S2/S4 helices and consequent VSD loss of function (*Wu et al., 2010*; *Figure 5B*). However, a simultaneous positively charged mutation in S2 provides a favorable electrostatic interacting partner for the S4 mutation and re-stabilizes the VSD (*Figure 5C*). Importantly, the electrostatic interactions between the S2/S4 helices are only energetically favorable when the mutated charge in S4 is aligned with the paired mutation in S2 (*Figure 5C*). The double mutations thus arrest the VSD conformation as dictated by the S4/S2 charge reversal mutation sites, yielding constitutively opened channels (*Wu et al., 2010*; *Zaydman et al., 2014*; *Papazian et al., 1995*; *Restier et al., 2008*; *Figure 5*).

We next looked for functional readouts to determine whether the arrested VSD conformations correspond to the intermediate or activated VSD states. We took advantage of the fact that KCNQ1 conducts current with distinct properties when its VSDs adopt either intermediate or activated states (*Zaydman et al., 2014*). The canonical open state associated with the activated VSD is referred to as the 'activated-open' (AO) state, while the distinct open state associated with the intermediate VSD is referred to as the 'intermediate-open' (IO) state. The AO and the IO states, and by inference the activated and intermediate VSD states, can be discriminated by two functional metrics. First, KCNQ1 co-expression with the accessory subunit KCNE1 selectively suppresses IO-state current by preventing pore opening when the VSD adopts the intermediate state (*Zaydman et al., 2014*). In addition, KCNE1 co-expression amplifies AO-state currents, in part by increasing single channel conductance (*Zaydman et al., 2014*; *Hou et al., 2017*), and possibly also by affecting VSD-pore coupling. *Figure 5D* summarizes KCNE1 regulation of the KCNQ1 IO-state and AO-state currents. These KCNE1 regulatory effects slow current activation (due to IO state suppression) and enhance current amplitude (due to AO state potentiation) of the WT KCNQ1 channels (*Figure 5E*; *Zaydman et al., 2014*). Second, the IO state is selectively inhibited by the KCNQ channel modulator XE991 compared to the AO state (*Zaydman et al., 2014*). Thus, current recordings in response to KCNE1 co-expression (*Figure 5*) and to XE991 exposure (*Figure 6*) allow us to test whether the S2-S4 interactions seen in the two structures correspond to the intermediate or activated VSD states.

We generated two classes of mutants designed to promote specific interactions based on the interacting residue pairs involving S2 and S4 observed in the differing NMR and cryo-EM VSD structures (*Figures 4* and *5A*). The first class of mutants was derived from the NMR-structure: E170R paired with R237E (E2R/R4E), and F167R with both Q234E and D202N (F0R/Q3E/D202N). The second class of mutants was based on interacting residue pairs observed in the cryo-EM structure (*Figures 4* and *5A*): F167R paired with H240E and D202N (F0R/H5E/D202N). An additional charge transfer center mutation D202N (in S3) was included along with the S4/S2 double charge reversal mutations F0R/Q3E and F0R/H5E. We had expected the double mutants F0R/Q3E and F0R/H5E to arrest S2-S4 registration, thereby yielding constitutively opened channels. However, both double mutants retain some levels of voltage-dependence in activation (*Figure 5—figure supplement 1*),

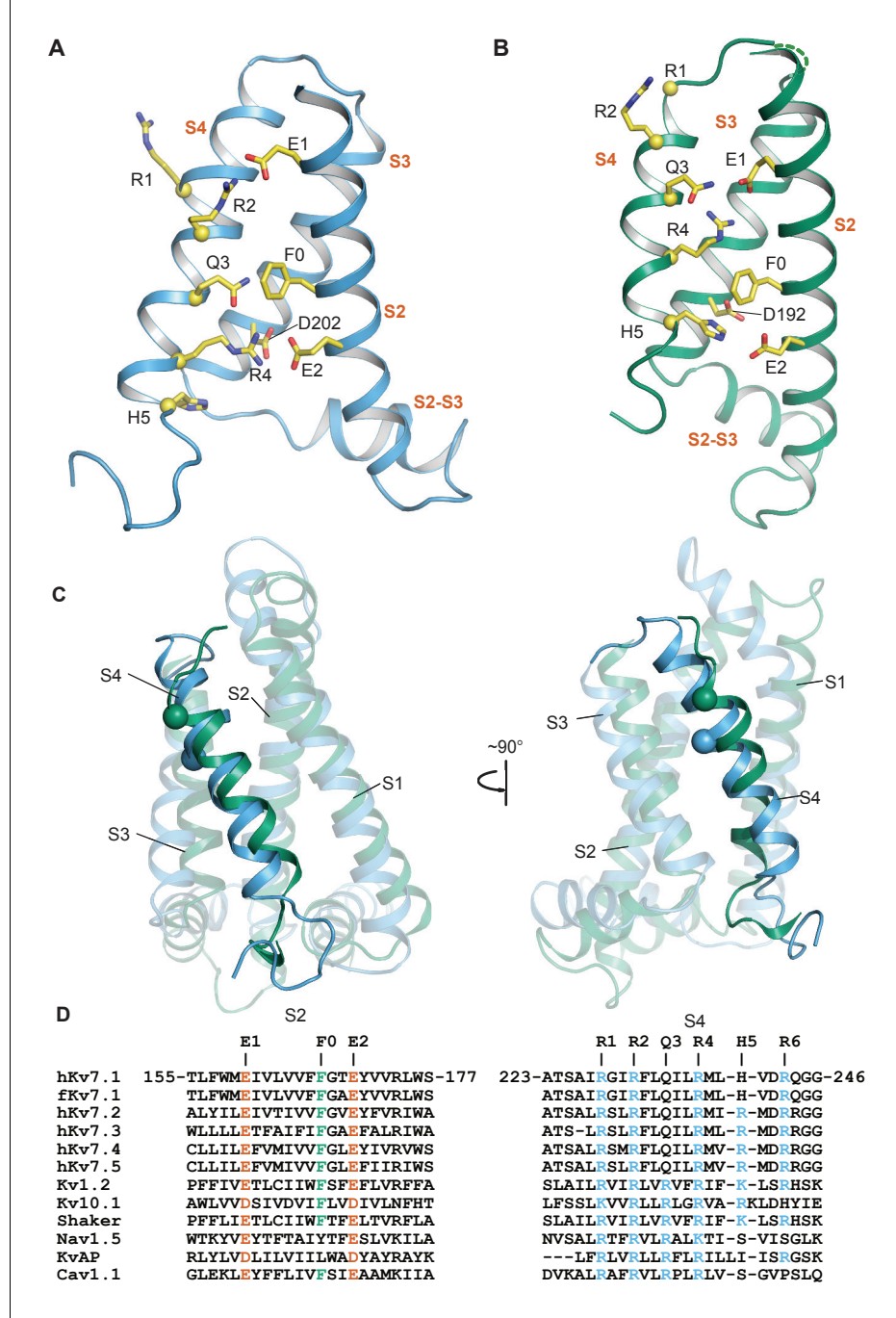

**Figure 3.** Comparison of intermediate and activated KCNQ1 VSD conformations. (**A**) Intermediate conformation of human KCNQ1 VSD (1st structure in the PDB 6MIE ensemble). (**B**) Activated conformation of the *Xenopus* KCNQ1 VSD (PDB 5VMS) (*Sun and MacKinnon, 2017*). In both panels, the Cα atoms of the S4 polar residues are shown as yellow spheres and the transmembrane helices are labeled in vermillion text. S0 and S1 are not shown to improve clarity of side chain interactions. (**C**) Overlay of the NMR (sky blue) and cryo-EM (bluish green) structures of the KCNQ1 VSD. All structural elements other than the S4 helix are semi-transparent. The Cα of the human residue G229 and the corresponding *Xenopus* residue G219 are shown as spheres. (**D**) Sequence alignments for S2 and S4 in the KCNQ/Kv7 family and select other voltage-gated ion channels.

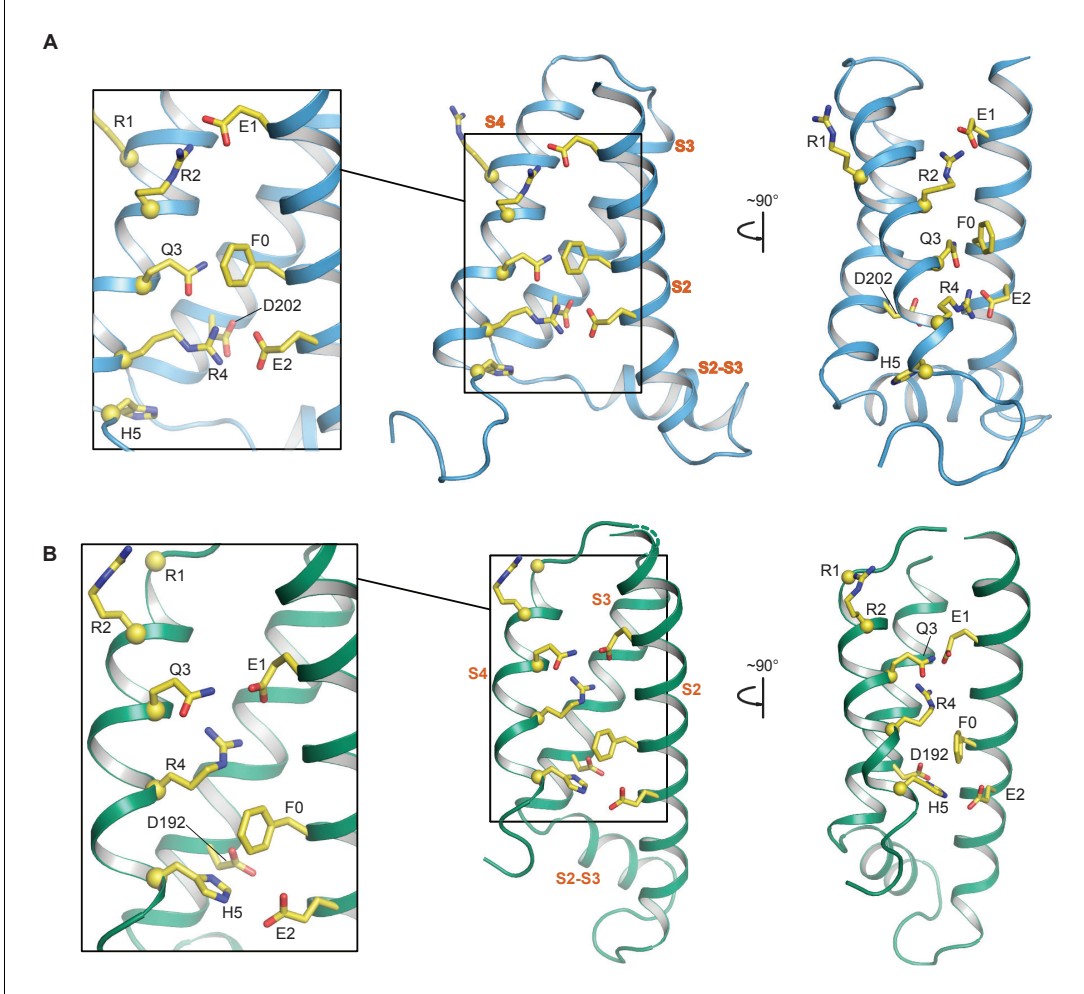

**Figure 4.** S2—S4 salt bridges/hydrogen bonds in the NMR and cryo-EM structures of the KCNQ1 VSD. (**A**) Intermediate state conformation of human KCNQ1 VSD (1st structure in the PDB 6MIE ensemble). Of particular note are the ionic interactions of E1-R2 and E2-R4, as well as, the close packing of Q3 and F0. (**B**) Activated state conformation of the *Xenopus* KCNQ1 VSD (PDB 5VMS). Note the gating charge residue pairings of E1-R4 and H5-E2. In all panels, the Cα atoms of the S4 polar residues are shown as yellow spheres and the transmembrane helices are labeled in vermillion text. S0 and S1 are not shown to improve clarity of side chain interactions.

suggesting that the double mutant only modestly stabilized the VSDs in their respective states. This voltage-dependence was eliminated upon the addition of the D202N mutation (*Figure 5F–H*), suggesting that D202 interfered with the ability of F0R/Q3E and F0R/H5E to arrest S2-S4 registration. This result also indicates that D202 is important for interacting with S4 gating charges during activation. As shown in *Figure 5F–H*, our designed KCNQ1 mutants (E2R/R4E, F0R/Q3E/D202N, and F0R/H5E/D202N) yielded constitutively open channel with minimal voltage dependence, consistent with the idea that the mutations strongly stabilize the VSDs in the intermediate or activated states by arresting the S2-S4 registration. Because the mutants were designed based on the NMR and cryo-EM VSD structures, these results also indicate that the VSDs in these mutant channels were arrested in the conformations corresponding the respective VSD structures. Next, we probed whether the VSD of these mutant channels corresponded to the functional intermediate and activated states by examining whether these KCNQ1 mutant channels were in the IO or AO state (*Figures 5* and *6*). We note that our experiments stabilized all four VSDs of KCNQ1 in the same conformation, thus we do not consider the pore conformation in the case of asymmetrical VSD states.

We first tested if these mutant channels exhibited the respective KCNE1 regulatory effect for the IO or AO state (*Figure 5D*). To test whether KCNE1 suppresses or enhances ionic currents of our mutants, we controlled for channel expression levels by injecting the mutant channel RNA in

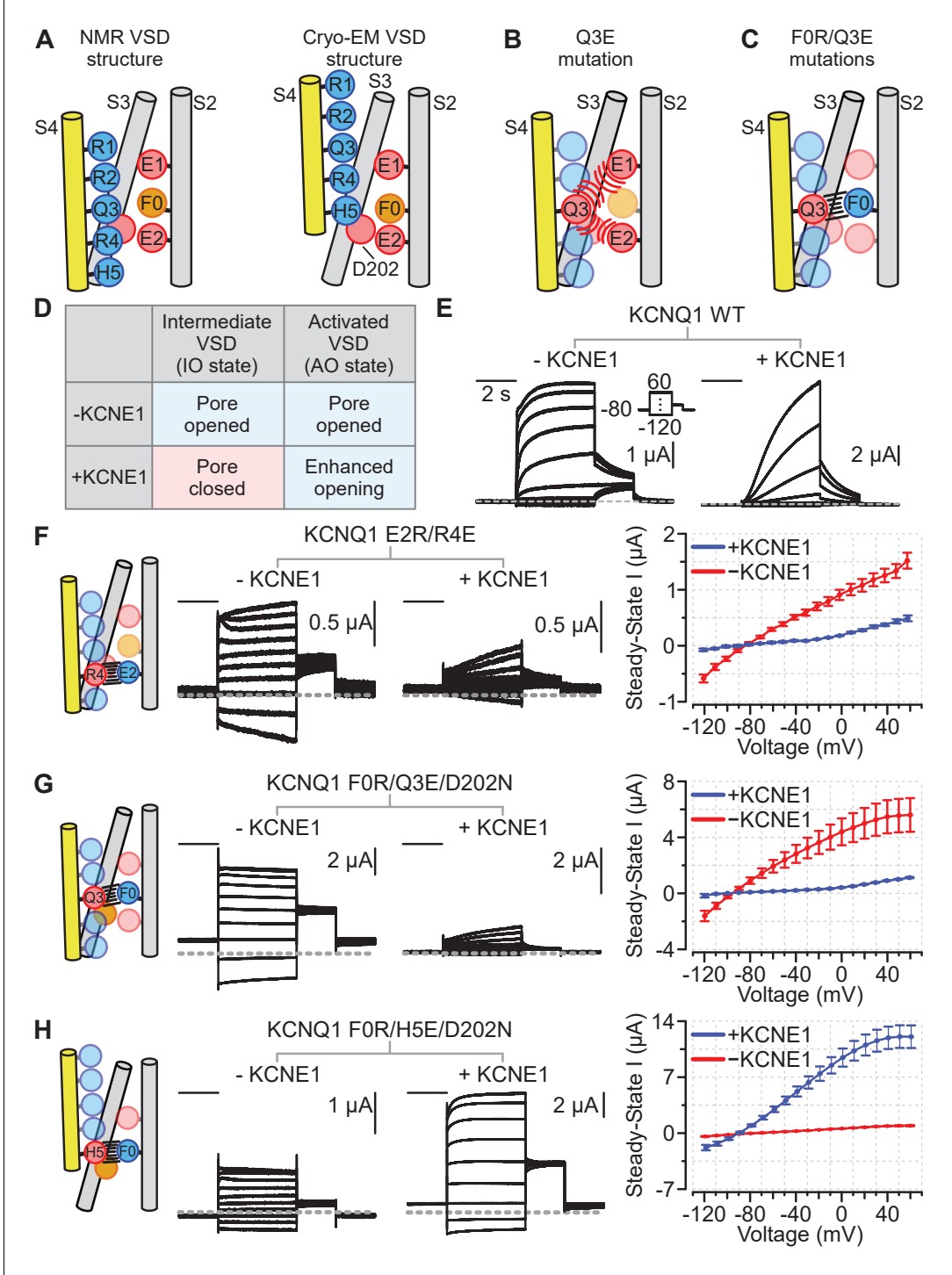

**Figure 5.** Schematics and electrophysiology data validating the intermediate and activated KCNQ1 VSD functional states utilizing auxiliary subunit KCNE1 regulation as a probe. Amino acid residue nomenclature: E2 = E170, R4 = R237, F0 = F167, Q3 = Q234, and H5 = H240. Numbering corresponds to the human KCNQ1 sequence. All error bars are ± SEM. All horizontal scale bars correspond to 2 s. (**A**) A cartoon schematic illustrating key S2, S3, and S4 residues interactions found in the NMR and cryo-EM VSD structures. Positive and polar gating residues on S4 (R1–H5) are colored blue, negative counter charges on S2 (E1, E2) and S3 (D202) are colored red, and the hydrophobic plug on S2 (F0) is colored orange. (**B**) A cartoon schematic displaying how the S4 charge-reversal mutation (Q3E) disrupts VSD function. The Q3E mutation creates electrostatic repulsion with the negative counter charges (E1, E2) and leads to VSD loss of function. (**C**) A cartoon schematic showing how the double charge-reversal mutations Q3E/F0R bias the VSD conformation. The double mutations ensure that electrostatic interactions between S2 and S4 are only favorable when the two mutation sites are in alignment (Q3E-F0R). (**D**) Table detailing KCNE1 effect on the KCNQ1 pore domain associated with the intermediate or activated

*Figure 5 continued on next page*

*Figure 5 continued*

VSD states based on prior studies. KCNE1 suppresses the IO state current by decreasing open probability, while enhancing the AO state current, in part by increasing unitary conductance (*Zaydman et al., 2014*; *Hou et al., 2017*; *Hou et al., 2019*; *Hou et al., 2020*). (E) Representative current recordings from the KCNQ1 channel without (left) and with (right) KCNE1 co-expression. The voltage protocol is shown in the inset and applies to all exemplars in this figure. (F–H) Left: Cartoon schematic of the double-charge reversal mutation and the predicted S2-S4 registry for the mutant tested. Middle: Exemplar currents for the mutant recorded with and without KCNE1 co-expression. Right: Average steady-state current vs. voltage (IV) curves for the respective mutants in the absence or presence of KCNE1 co-expression. The inset in panel E shows the voltage protocol. n = 5 (F), 5 (G), 6 (H). Currents were collected with 10 mV interval, but examples are shown with 20 mV interval for clarity.

The online version of this article includes the following source data, source code and figure supplement(s) for figure 5:

**Source data 1.** Excel file with numerical data used for *Figure 5*.
**Figure supplement 1.** Electrophysiology results for KCNQ1 F0R single and double mutants with and without KCNE1 co-expression, and with XE991 exposure.
**Figure supplement 1—source code 1.** MATLAB script that takes in tail current data obtained from PatchMaster program (see Key Resources Table), fits the data with a Boltzmann equation, and outputs the best fit parameters.
**Figure supplement 1—source data 1.** Excel file with numerical data used for *Figure 5—figure supplement 1*.

*Xenopus* oocytes with and without KCNE1 RNA co-injection on the same day. We then made recording using the same mutant with and without KCNE1 RNA co-injection on the same day post-injection (see Materials and methods). Because channel expression was not controlled across mutant channels (e.g. E2R/R4E vs. F0R/Q3E/D202N), we did not interpret current amplitudes between different mutants. The mutants E2R/R4E and F0R/Q3E/D202N resulted in constitutive opening of the channel (*Figure 5F,G*), which is consistent with stabilization of the intermediate VSD state by interactions between E2-R4 and F0-Q3. Consistently, KCNE1 co-expression strongly suppressed currents conducted by both mutants as shown by the current exemplar and average I-V curves (*Figure 5F,G*), confirming that interactions of these mutant residues stabilize the IO state (*Figure 5D*). The mutant F0R/H5E/D202N also resulted in constitutively open channels similar to E2R/R4E and F0R/Q3E/D202N (*Figure 5H*). However, in contrast to the prior two mutants, KCNE1 co-expression greatly enhanced F0R/H5E/D202N current as illustrated by the exemplar currents (note scale bars) and average I-V curves (*Figure 5H*), confirming the hypothesis that F0-H5 interaction stabilizes the AO state (*Figure 5D*).

We also found that KCNE1 co-expression could distinguish between the double mutants F0R/Q3E and F0R/H5E that retain some levels of voltage dependence. First, KCNE1 co-expression suppressed current amplitudes of the F0R/Q3E mutant (Figure 5—figure supplement 1C) but greatly enhanced current amplitudes of the F0R/H5E mutant (Figure 5—figure supplement 1E). Second, KCNE1 co-expression right-shifted the conductance-voltage (G-V) relation of F0R/Q3E mutant significantly more than that of F0R/H5E (Figure 5—figure supplement 1C,E), indicating that F0R/Q3E favors the IO state and F0R/H5E promotes the AO state. These results are consistent with those of the triple mutants F0R/Q3E/D202N and F0R/H5E/D202N (*Figure 5G,H*). Altogether, the KCNE1 co-expression experiments unambiguously indicate that the KCNQ1 mutants designed according to the NMR and cryo-EM VSD structures occupy the IO and the AO states, respectively (*Figure 5F–H*, *Figure 5—figure supplement 1*). These KCNE1 co-expression results thus indicate that the NMR structure of the KCNQ1 VSD corresponds to the intermediate state, while the cryo-EM structure of the KCNQ1 VSD populates the activated state.

We next examined whether XE991 pharmacology might consistently identify these mutant channels at the IO or AO state. Previous studies found that the KCNQ modulator XE991 at 5 µM preferentially inhibits the IO-state current over the AO-state current, as summarized in Figure 6A; *Zaydman et al. (2014)*. We started by probing the effect of XE991 on the KCNQ1 E2R/R4E mutant. We first recorded oocytes expressing the KCNQ1 E2R/R4E channels in control ND96 solution. The channels were held at −20 mV and pulsed to +40 mV for 4 s (test pulse) and −40 mV for 2 s (tail pulse) every 20 s. *Figure 6B* visualizes exemplar E2R/R4E mutant current amplitudes at the end of the 40 mV test pulse over time throughout the experiment. After the E2R/R4E current amplitude reached a stable level under control conditions (*Figure 6B*, black arrow and current trace), we applied 5 µM XE991 and continued recording until the current amplitude reached steady state (*Figure 6B*, red arrow and current trace). The E2R/R4E current amplitude was relatively small at around 1 to 1.5 µA (*Figures 5F* and *6B*). The smaller current amplitude may lead to poor estimation

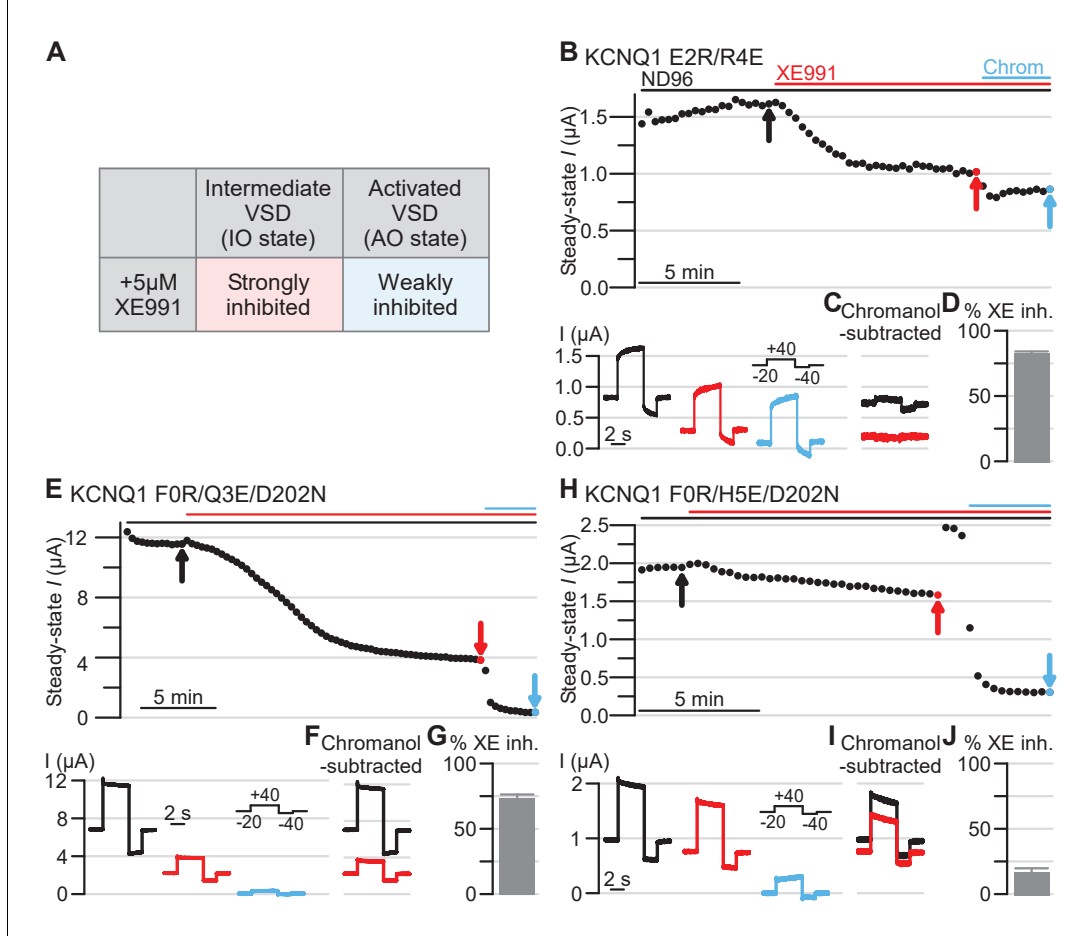

**Figure 6.** Electrophysiology validating the intermediate and activated KCNQ1 VSD functional states utilizing XE991 pharmacology as a probe. Amino acid residue nomenclature and numbering and error bars are similar as in *Figure 5*. (A) Table outlining 5 μM XE991 effect on KCNQ1 IO and AO state currents based on prior studies (*Zaydman et al., 2014*). (B) Top: Exemplar diary plots of E2R/R4E drug studies demonstrating current amplitude over time. Cells were held at −20 mV and pulsed to +40 mV for 4 s and −40 mV for 2 s every 20 s. Each point shows the steady-state current amplitude at the end of the 4 s +40 mV test pulse. Cells were recorded in ND96 solution and the top bars indicate application of 5 μM XE991 (red) and 150 μM chromanol 293B (blue). Scale bar indicates 5 min. Bottom: Current traces for the E2R/R4E mutant in control ND96 solution (black), in solution containing 5 μM XE991 (red), and in solution containing 150 μM chromanol 293B (blue). The arrows in the diary plot indicate respective traces shown. Note that because the holding potential was −20 mV and the mutant channels are constitutively open, non-zero currents were observed before the test pulse. (C) Chromanol-subtracted E2R/R4E currents under control (black) and 5 μM XE991 (red) conditions for the traces shown in panel B. The chromanol-subtracted currents were calculated by subtracting current after chromanol application (blue, panel B) from the control current (black, panel B) and the current after XE991 application (red, panel B). Percent E2R/R4E current inhibition by XE991 was calculated from the chromanol-subtracted currents using the ratio between the steady-state current amplitude under XE991 and control conditions (see Materials and methods). (D) Average percent inhibition of the E2R/R4E currents by 5 μM XE991, as quantified by the chromanol-subtracted currents (n = 6). Error bar indicates SEM and applies to all error bars in this figure. (E–J) Same as panels B-D, but showing results for KCNQ1 F0R/Q3E/D202N and F0R/H5E/D202N mutants (n = 6 for both mutants).

The online version of this article includes the following source data for figure 6:

**Source data 1.** Excel file with numerical data used for *Figure 6*.

of E2R/R4E current inhibition by XE991 due to endogenous oocyte current contamination. We therefore subsequently applied 150 μM chromanol 293B, a selective I_Ks and KCNQ1 blocker, to the bath containing 5 μM XE991 (*Figure 6B*). We calculated XE911 inhibition of E2R/R4E currents by first subtracting the current after chromanol application (*Figure 6B*, blue arrow and current trace) from the currents under control and 5 μM XE991 conditions (*Figure 6B*, black and red currents respectively). As plotted in *Figure 6C*, the resulting chromanol-subtracted currents represent XE991 inhibition of E2R/R4E currents without endogenous current contamination. The percent XE991 inhibition was

calculated using the ratio of the chromanol-subtracted current amplitudes of the control and XE991 conditions (see Materials and methods). As illustrated by the exemplar and the average inhibition bar plot (*Figure 6C*, black to red currents, and 6D), XE991 robustly inhibited ~80% of E2R/R4E currents, suggesting that the E2R/R4E mutant stabilizes the IO state (*Figure 6A*).

Likewise, we found that 5 μM XE991 also significantly inhibited F0R/Q3E/D202N mutant currents as demonstrated in *Figure 6E–F*, with an average inhibition of ~75% (*Figure 6G*). This result confirms that the F0R/Q3E/D202N mutant stabilizes the IO state, similar to the E2R/R4E mutant. In contrast, 5 μM XE991 was far less effective at inhibiting F0R/H5E/D202N currents, as shown in the exemplar and average bar plots, with an average inhibition ~20% (*Figure 6H–J*). This result indicates that the F0R/H5E/D202N mutant promotes the AO state and thus conducts current resistant to XE991 inhibition (*Figure 6A*). Moreover, the XE991 pharmacology experiments revealed consistent findings with the double mutants F0R/Q3E and F0R/H5E, in which the F0R/Q3E mutant was robustly inhibited by 5 μM XE991 inhibition while the F0R/H5E was insensitive to 5 μM XE991 (Figure 5—figure supplement 1F–J). These double mutant results agree with data from the triple mutants F0R/Q3E/D202N and F0R/H5E/D202N (*Figure 6E–J*), further supporting the notion that F0-Q3 and F0-H5 interactions are found in IO and the AO states, respectively.

Critically, data from these XE991 experiments corroborate the IO- and AO-state discrimination between VSD mutants deduced from the KCNE1 co-expression data (*Figure 5*). Taken together, these two sets of results (*Figures 5* and *6*, *Figure 5—figure supplement 1*) strongly suggest that E2-R4 and F0-Q3 are interactions found in the KCNQ1 VSD intermediate state, while the F0-H5 interaction is present in the activated state. These data validate that the NMR VSD structure represents a conformation that corresponds to the stable intermediate state of the VSD during voltage-dependent activation, while the VSD in the cryo-EM structure represents the activated state.

## KCNQ1 VSD activation motion from the intermediate to the activated state

Comparison of the two VSD structures (*Figure 3C*) reveals a pronounced S4 helix movement relative to the rest of the VSD upon transition from the intermediate to the activated state, with a ~ 5.4 Å translation of S4 toward the extracellular direction accompanied by unraveling of the N-terminal end of this helix, perhaps as a result of its transition into a well-hydrated extracellular environment (*Figure 3C*). Consequently, the S4 helix of the intermediate state is longer by two additional turns between V221 and G229, suggesting a simultaneous loss of the secondary structure in the N-terminal portion of S4 during the transition from the intermediate to activated state. Other helices, especially the extracellular half of S3, undergo only modest translations, as evident in an overlay of the two structures (*Figure 3C*). In both structures, the S4 charges form ion pairs with E1, E2, and D202, but with different registrations. Our functional studies demonstrated that these ion pairs provide much of the energy to stabilize the VSD in the intermediate and activated states during voltage-dependent activation (*Figures 5* and *6*), thereby delimiting the trajectory of VSD motions during the intermediate-to-activated state transition.

## Physiological role of the intermediate state of the KCNQ1 voltage sensor

Our study so far presents a structure of the KCNQ1 VSD and provides functional evidence that the structure represents a stable intermediate conformation along the KCNQ1 VSD activation pathway. In our functional validation, we extensively utilized the distinct KCNQ1 intermediate conductive IO state as a readout for the intermediate VSD state. However, little is known regarding the physiological role of this conductive IO state. Both this report and prior studies indicate that the auxiliary subunit KCNE1 suppresses the IO state (*Zaydman et al., 2014*; *Figure 5*). In cardiac myocytes, KCNQ1 is known to complex with KCNE1 to generate the $I_{Ks}$ current required for cardiac action potential termination (*Barhanin et al., 1996*; *Sanguinetti et al., 1996*; *Chiamvimonvat et al., 2017*; *Keating and Sanguinetti, 2001*). The KCNQ1 IO state thus likely minimally impacts normal cardiac physiology. What role might the KCNQ1 intermediate VSD state and its associated conductive IO state play in normal physiology? To answer this question, we look beyond cardiac tissues.

KCNQ1 is unusual in that its functional properties vary profoundly in association with different tissue-specific KCNE accessory proteins (*Liin et al., 2015*; *McCrossan and Abbott, 2004*). In epithelial

cells, KCNQ1 associates with KCNE3 to form a 'leak' potassium current essential for epithelial ion homeostasis (*Abbott, 2016*; *Julio-Kalajzić et al., 2018*; *Schroeder et al., 2000*; *Kroncke et al., 2016*; *Preston et al., 2010*). KCNE3 renders KCNQ1 current constitutively active in physiological voltage ranges as shown in *Figure 7A*, by contrast to the time- and voltage-dependent cardiac $I_{Ks}$ (KCNQ1/KCNE1). These strikingly distinct KCNQ1 currents fit their respective tissue-specific needs. On one hand, cardiac physiology demands the $I_{Ks}$ channel to conduct late during an action potential. This explains why KCNE1 suppresses the IO state and restricts KCNQ1 pore opening to VSD transition into the activated conformation (*Zaydman et al., 2014*). On the other hand, epithelial physiology requires KCNQ1+KCNE3 to conduct over wide-ranging voltages, including more hyperpolarized potentials where the intermediate VSD state is energetically favored over the activated VSD state. We therefore hypothesized that KCNE3 may render KCNQ1 constitutively active in part by utilizing the intermediate VSD conformation and the IO state.

To examine whether the KCNQ1/KCNE3 complex conducts significant current with the IO state, we undertook voltage-clamp fluorometry (VCF) experiments. VCF tracks KCNQ1 VSD transitions by a labeled fluorophore attached to the S3-S4 linker, which changes fluorescence emission during voltage-dependent activation (*Barro-Soria et al., 2014*; *Zaydman et al., 2014*; *Osteen et al., 2012*; *Barro-Soria et al., 2015*; *Barro-Soria et al., 2017*; *Nakajo, 2019*). The KCNQ1 fluorescence-voltage (F-V) relation exhibits two components that can be well-fit by a double Boltzmann function ($F_1$ and $F_2$ in *Figure 7A,B*), which correspond to VSD sequential transitions from resting to intermediate ($F_1$) and from intermediate to activated ($F_2$) states (*Zaydman et al., 2014*; *Hou et al., 2017*). Selective regulation of distinct VSD transitions can be inferred from changes to $F_1$ and $F_2$. Comparison of the G-V relation with the F-V relation provides insight into IO vs. AO state regulation. For example, it has been shown that KCNE1 co-expression specifically causes a hyperpolarized shift in the $F_1$ component of the F-V relation but depolarizes the G-V curve to follow the $F_2$-V relation (*Figure 7A*; *Barro-Soria et al., 2014*; *Zaydman et al., 2014*; *Hou et al., 2017*). Our previous study indicated that this phenomenon derives from a mechanism in which KCNE1 eliminates the IO state by preventing pore opening when the VSD is in the intermediate state, such that $I_{Ks}$ represents conductance only in the AO state (*Zaydman et al., 2014*; *Hou et al., 2017*). KCNE3 co-expression has also been demonstrated to induce a hyperpolarizing shift in the F-V relation, suggesting that KCNE3 promotes channel opening by shifting the voltage dependence of VSD activation (*Barro-Soria et al., 2015*; *Barro-Soria et al., 2017*).

A careful inspection of VCF measurements reveals that KCNE3, like KCNE1, also specifically hyperpolarizes the $F_1$ component while having little effect on the $F_2$ component (*Figure 7A,B*, see also [*Barro-Soria et al., 2015*; *Barro-Soria et al., 2017*]). However, unlike KCNE1, KCNE3 shifted the G-V curve in a hyperpolarizing direction to follow the $F_1$-V curve (*Figure 7A*). This led us to hypothesize that unlike KCNE1, KCNE3 association does not prevent pore opening when the VSD adopts the intermediate state. This was tested by co-expressing KCNE3 with KCNQ1-F351A, a mutant known to be non-conductive when the VSD occupies the intermediate state (*Zaydman et al., 2014*; *Hou et al., 2017*). The KCNQ1-F351A/KCNE3 channel complex maintained a hyperpolarized $F_1$ component, but the G-V relationship changed to track the $F_2$-V curve, confirming that the currents observed for wild type KCNQ1/KCNE3 at negative voltages are conducted by the IO state (*Figure 7B*). KCNE3 also preserves the AO state as shown by two observations. First, residual pore opening observed in KCNQ1-F351A/KCNE3 indicates that the AO state is intact, as the KCNQ1-F351A channels cannot conduct in the IO state (*Figure 7B*). Second, subtraction of the instantaneous current from KCNQ1-WT/KCNE3 current reveals a time- and voltage-dependent current which tracks the $F_2$ component (*Figure 7C*, GV$_2$ curve), suggesting that the time-dependent fraction of KCNQ1-WT/KCNE3 channels conduct at the AO state at high voltages. Taken together, these results demonstrate that KCNQ1/KCNE3 conducts with both the IO and the AO states. However, our VCF data indicate that the intermediate VSD state is more favorably occupy at hyperpolarized voltages (*Figure 7A*), suggesting that the IO state may significantly contribute to physiological KCNQ1/KCNE3 currents.

To further examine whether KCNQ1/KCNE3 preferentially conducts at the IO state, we compared XE991 inhibition of KCNQ1/KCNE3 and KCNQ1/KCNE1. *Figure 7D* shows the overlays of KCNQ1, KCNQ1/KCNE1, and KCNQ1/KCNE3 current traces stabilized in control ND96 solutions (black) and solutions containing 5 µM XE991 (gray, blue, red). Although the KCNQ1 AO-state current is resistant to 5 µM XE991, KCNE1 was previously shown to sensitize the AO state to permit some XE991

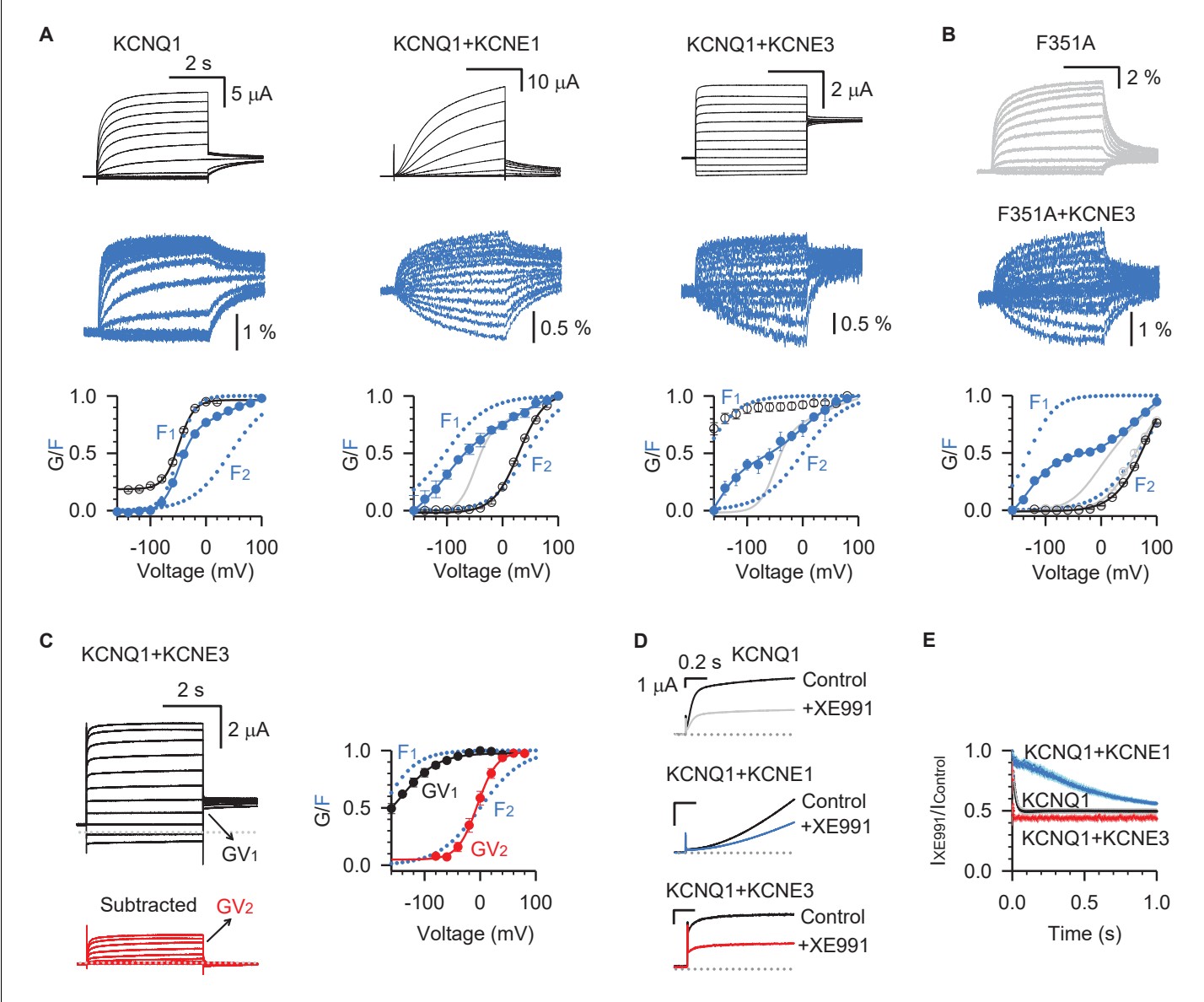

**Figure 7.** KCNE3 shifts the voltage dependence of the intermediate-open state to render its conductance relevant under physiological conditions. (A) Voltage-clamp fluorometry recordings for pseudo-WT (C214A/G219C/C331A) KCNQ1 (left), KCNQ1+KCNE1 (middle), and KCNQ1+KCNE3 (right). Current (black) and fluorescence (blue) were recorded with voltages from −160 mV to +100 mV in 20 mV increments and then back to −40 mV. The bottom panels are the G–V (black) and F–V (blue) relationships with $F_1$ and $F_2$ components (dotted lines). The F–V relationship for KCNQ1 (gray) is also shown with KCNQ1+KCNE1 and KCNQ1+KCNE3 for comparison. (B) VCF recordings for KCNQ1-F351A in the presence (blue) and absence (gray) of KCNE3. The bottom of this panel shows the G–V and F–V relationships for KCNQ1-F351A (gray) and KCNQ1-F351A+KCNE3 (blue). (C) Left: WT KCNQ1+KCNE3 currents (black) with voltages from −120 mV to +80 mV in 20 mV increments and back to −40 mV to test the tail current. The red traces are total KCNQ1+KCNE3 currents with the instantaneous current subtracted, which show typical time- and voltage-dependent activation. Right: The two steps of KCNQ1+KCNE3 voltage sensor activation, $F_1$ and $F_2$, are shown in blue, while transitions of the two conductive states are shown in black ($GV_1$) and red ($GV_2$). (D) Overlays of stabilized currents in control solutions (black traces for all channels) and after applying 5 µM XE991 for KCNQ1 (gray), KCNQ1+KCNE1 (blue), and KCNQ1+KCNE3 (red). (E) Time-dependent inhibition of 5 µM XE991 of KCNQ1 (black), KCNQ1+KCNE1 (blue), and KCNQ1+KCNE3 (red). Time-dependent inhibition was calculated by dividing the stabilized currents in solutions containing 5 µM XE991 by that of control at the same time point. X-axis is the time after the start of the +40 mV test pulse. In all cases, n ≥ 3.

The online version of this article includes the following source data for figure 7:

**Source data 1.** Excel file with numerical data used for *Figure 7*.

inhibition (*Zaydman et al., 2014*). Still, KCNE1 and KCNE3 induced distinct XE991 blocking kinetics of KCNQ1 currents during each pulse of channel activation. *Figure 7E* illustrates the time-dependent inhibition of XE991 by plotting the current amplitudes under 5 μM XE991 conditions divided by that of the control, with the x-axis indicating time after the voltage was stepped to +40 mV. KCNQ1/KCNE1 current featured much slower XE991 time-dependent block compared to KCNQ1 alone, suggesting that KCNQ1/KCNE1 primarily conducts at the AO state. By contrast, the KCNQ1/KCNE3 current exhibited rapid XE991 inhibition with kinetics faster than KCNQ1 alone. This result suggests that KCNQ1/KCNE3 preferentially conducts IO-state current, which is sensitive to XE991. Taken together, our results demonstrate that in the physiological voltage range (−80 to +50 mV) the KCNQ1/KCNE3 complex preferentially populates the conductive IO state over the AO state, as promoted by the occupancy of the VSDs in the intermediate state (*Figure 7A*).

## Discussion

This study took advantage of the unique ability of LMPG micelles to stabilize the intermediate state of the human KCNQ1 VSD, leading to determination of its structure using a robust approach combining NMR spectroscopy with data-restrained MD. The intermediate state VSD structure complements both the previous activated state *Xenopus* KCNQ1 VSD structure in DDM micelles determined using cryo-EM (*Sun and MacKinnon, 2017*) and homology models for human KCNQ1 in both the fully activated and resting states (*Kuenze et al., 2019*). Both the new NMR structure and the cryo-EM VSD structures are consistent with structure-function results from previous studies of KCNQ1 and were also rigorously validated in this work by additional electrophysiological studies. Determination and validation of the KCNQ1 VSD structures in the intermediate and activated states furnishes critical information for understanding the voltage-dependent activation of KCNQ1 and interactions with KCNE accessory subunits. The intermediate and activated VSD structures are seen to be distinctly different, which is as expected because they are coupled to channel conductance in ways that exhibit very different sensitivities to KCNE1 and drug binding. We acknowledge that the intermediate state VSD structure was determined in isolation from the rest of the channel and that future studies will be required to elucidate exactly how it is coupled to the pore to promote the IO conductance state.

Even with the limitation indicated immediately above, the intermediate state VSD structure can be used in conjunction with the cryo-EM structure of KCNQ1 (*Sun and MacKinnon, 2017*) and with available models for both resting and fully activated human KCNQ1 (*Kuenze et al., 2019*) to illuminate KCNQ1 pharmacology, physiology, and how some of the dozens of known LQTS disease mutations located in the VSD result in channel loss of function or dysfunction (*Tobelaim et al., 2017*; *Huang et al., 2018*; *Vanoye et al., 2018*). Moreover, the KCNQ1 VSD intermediate state structure can be used as a template to model the intermediate state voltage sensors of other voltage-gated channels, such as the Shaker potassium channel (*Bezanilla et al., 1994*; *Zagotta et al., 1994*; *Baker et al., 1998*; *Jensen et al., 2012*; *Lacroix et al., 2012*), and voltage-gated Na channels (*DeCaen et al., 2008*), which are believed to activate in a step-wise manner with S4 gating charges sequentially pairing with the charge transfer center and other acidic residues.

This work utilizes KCNQ1's unique intermediate conductance to provide functional electrophysiology evidence that the VSD structure presented here truly represents the intermediate state conformation. We presented functional validation based on two independent metrics, auxiliary subunit KCNE1 regulation and XE991 pharmacology (*Figures 5* and *6*). While the precise mechanism by which KCNE1 suppresses the IO state and enhances the AO state remains unclear, the fact that KCNE1 co-expression unambiguously discriminates channel mutants designed to stabilize the NMR VSD structure vs. the cryo-EM VSD structure constitutes compelling evidence that the two structures represent distinct functional KCNQ1 VSD states. Moreover, pharmacological studies of the same mutants by XE991 fully corroborate KCNE1 co-expression results, indicating that our functional validation data are robust and not dependent on any single functional result. By contrast, use of functional measurements to illuminate the properties of the intermediate state versus those of the activated state would be very challenging for $K_V$ channels that do populate an intermediate VSD state, but for which the pore is thought to only conduct current when its VSDs occupy the activated state, such as the Shaker channel. In the case of the Shaker channel, functional electrophysiology would be unable to distinguish the intermediate state vs. the resting state of the VSD, as both states

yield no ionic currents. KCNQ1 thus represents a particularly suitable platform for structure-function study of the intermediate state in $K_V$ VSDs.

In addition to utilizing KCNQ1's intermediate conductance as a functional probe, this work also suggests a possible physiological role for the KCNQ1 intermediate conductance. We demonstrated that under physiological conditions where KCNQ1 is paired with KCNE3 as an accessory subunit, the channel is maximally conductive over a wide range of transmembrane voltages. Depending on the transmembrane electrical potential, conductance may be a mix of the IO and AO states, as we observe that KCNE3 preserves both open states (*Figure 7*). Two cryo-EM structures of the full-length KCNQ1/KCNE3/calmodulin complex with and without $PIP_2$ were published during the revision of this manuscript (*Sun and MacKinnon, 2020*). In both new structures, the VSDs adopt the fully activated conformation at 0 mV membrane potential (*Sun and MacKinnon, 2020*), consistent with our finding that KCNQ1/KCNE3 complex conducts some AO-state current at depolarized voltages. Nevertheless, our VCF and XE991 results demonstrate that that KCNE3 may more favorably stabilize the KCNQ1 VSDs in the intermediate state at more hyperpolarized voltages (−150 to −40 mV, *Figure 7A* $F_1$ curve for KCNQ1/KCNE3). The primary function of the KCNQ1/KCNE3 complex is to serve as a leak channel in epithelial cells (*Abbott, 2016*; *Julio-Kalajzić et al., 2018*; *Schroeder et al., 2000*; *Kroncke et al., 2016*; *Preston et al., 2010*). As epithelial cells are non-excitable cells and not physiologically subject to the highly depolarizing membrane potentials required to activate substantial AO state conductance, the IO state conductance likely constitutes the bulk of the physiological KCNQ1/KCNE3 leak $K^+$ current in epithelial cells. Thus, the IO state is not merely a transient state on the pathway from resting to the AO state. Rather, the data suggests that IO is itself an essential conductive state under some physiological conditions. KCNE1 and KCNE3 therefore promote different open states to confer exceptional functional versatility, uniquely tailoring KCNQ1 so that it can satisfy distinct physiological needs in different cell types and tissues.

## Materials and methods

**Key resources table**

| Reagent type (species) or resource | Designation | Source or reference | Identifiers | Additional information |
|---|---|---|---|---|
| Gene (human) | KCNQ1 | HUGO Gene Nomenclature Committee (HGNC) | Gene ID: 3784; HGNC:6294 | |
| Gene (human) | KCNE1 | HUGO Gene Nomenclature Committee (HGNC) | Gene ID: 3753; HGNC:6240 | |
| Gene (human) | KCNE3 | HUGO Gene Nomenclature Committee (HGNC) | Gene ID: 10008; HGNC:6243 | |
| Strain, strain background (*Escherichia coli*) | CT19 transaminase deficient strain | Dr. David Waugh, US National Cancer Institute PMID: 8914274 | | Used for special isotopic labeling of the KCNQ1 VSD for use in NMR studies. |
| Strain, strain background (*Escherichia coli*) | Rosetta/ C43(DE3) | Sigma-Aldrich | Catalog number 70954 | Used for uniform isotopic labeling of KCNQ1 and amino acid-specific labeling for NMR. |
| Recombinant DNA reagent | pET16b expression plasmid encoding tagged human KCNQ1-(100-249) | PMID: 24606221 | | Used to express the human KCNQ1 VSD for NMR structural studies |
| Biological sample (include species here) | *Xenopus* oocytes (*Xenopus laevis*, female) | This paper | RRID:XEP_Xla | *Xenopus laevis* purchased from Nasco, Fort Atkinson, WI |
| Recombinant DNA reagent | pcDNA3.1 encoding human KCNQ1 or KCNE1 | This paper | RRID:Addgene_111452 | For site-directed mutagenesis |

*Continued on next page*

*Continued*

| Reagent type (species) or resource | Designation | Source or reference | Identifiers | Additional information |
|---|---|---|---|---|
| Commercial assay or kit | mMessage T7 polymerase kit | Applied Biosystems-Thermo Fisher Scientific | AM1344 | |
| Chemical compound, drug | XE991 | Millipore Sigma | CAS #: 122955-42-4 | |
| Chemical compound, drug | Chromanol 293B | Millipore Sigma | CAS #: 163163-23-3 | |
| Chemical compound, drug | Alexa Fluor 488 C5 maleimide | Molecular Probes, Eugene, OR | Catalog #: A10254 | |
| Software | Topspin 3.2 | Bruker (Scientific Instruments Company) | RRID:SCR_014227 | NMR data collection and processing. |
| Software | MDD and qMDD interface | URL: mddnmr.spektrino.com/ PMID: 21161328 | | NMR data processing |
| Software | NMRFAM-SPARKY | PMID: 25505092 URL: https://nmrfam.wisc.edu/nmrfam-sparky-distribution/ | | NMR data analysis and resonance assignments. |
| Software | TALOS-N | PMID:25502373 URL: spin.niddk.nih.gov/bax/software/TALOS-N/ | | Determination of secondary structure from NMR chemical shift data. |
| Software | CHARMM-GUI | PMID: 25130509 URL: www.charmm-gui.org/ | | Preparation of starting structures of the KCNQ1 VSD in lipid bilayers for MD restrained MD simulations. |
| Software | XPLOR-NIH | PMID: 28766807 URL: https://nmr.cit.nih.gov/xplor-nih/ | | Structure calculations using NMR data restraints |
| Software | GPU-accelerated AMBER 16 | URL: https://ambermd.org/ PMID: 16200636 | | Program for execution of MD simulations. |
| Software | Lipid 17 AMBER | PMID: 24803855 URL: https://ambermd.org/AmberModels.php | | Force field used for MD simulations. |
| Software | CPPTRAJ | PMID: 26583988 URL: https://amber-md.github.io/ | | Analysis of MD trajectories following simulations. |
| Software | PatchMaster | HEKA | RRID:SCR_000034 | Electrophysiology data collection |
| Software | IGOR | Wavemetrics, Lake Oswego, OR | RRID:SCR_000325 | Electrophysiology data analysis |
| Software | Clampfit | Axon Instruments, Sunnyvale, CA | RRID:SCR_011323 | Electrophysiology data analysis |
| Software | Sigmaplot | SPSS, San Jose, CA | RRID:SCR_003210 | Electrophysiology data analysis and visualization |
| Software | MATLAB | MathWorks, MA | RRID:SCR_001622 | Electrophysiology data analysis |

*Continued*

| Reagent type (species) or resource | Designation | Source or reference | Identifiers | Additional information |
|---|---|---|---|---|
| Sequence-based reagent | For site-directed mutagenesis | This paper | PCR primers | PCR primers seq for mutations made in this study (each mutation utilized two primers: b and c). E170R-b:cacgtacCTGgtcccgaagaacaccac; E170R-c: cgggacCAGgtacgtggtccgcctc; R237E-b: gcatcTCcaggatctgcaggaag; R237E-c:cagatcctgGAgatgctacacgtcgac F167R-b: ccgtcccgCGgaacaccaccagcac; F167R-c: gtgttcCGcgggacggagtacg Q234E-b:ggatctCcaggaagcggatgccc; Q234E-c: catccgcttcctgGagatcctgaggatgcta H240E-b: cggtcgacCTCtagcatcctcaggatc H240E-c: gctaGAGgtcgaccgccaggg D202N-b: cgatgaggtTAatgatggaaatgggcttc D202N-c: ccatcatTAacctcatcgtggtcgtg F351A-b: ggcaGCGcccgagccaagaatcc F351A-c: gctcgggCGCtgccctgaaggtgcag C214A-b: cttggaTcccacCGCgaggaccacca C214A-c: cGCGgtgggAtccaaggggcaggtg G219C-b: cctgaCacttggaTcccacCGC G219C-c: ggAtccaagtGtcaggtgtttgccacg C331A-b: gacagagaaTGCggaggcgatggtcttc C331A-c: ctccGCAttctctgtctttgccatc |

## Constructs, mutagenesis, and expression

Point mutations of the KCNQ1 channel were engineered using overlap extension and high-fidelity PCR. Each mutation was verified by DNA sequencing. The cRNA of mutants was synthesized using the mMessage T7 polymerase kit (Applied Biosystems-Thermo Fisher Scientific).

The human KCNQ1 (GenBank accession number AF000571) VSD was cloned into a pET16b expression vector as previously described (*Peng et al., 2014*). The pET16b expression construct included an N-terminal His tag of the sequence MGHHHHHHG followed by KCNQ1 residues 100–249. Single amino acid mutations were generated by QuikChange site-directed mutagenesis and verified by sequencing to confirm the presence of the desired codon substitutions. For expression, the *E. coli* strain C43(DE3) harboring the pRARE plasmid (encoding rare codon tRNAs) was transformed with the KCNQ1-VSD pET16b vector. Transformants were cultured overnight in 3 mL of LB media containing 100 µg/mL ampicillin and 30 µg/mL chloramphenicol at 37°C. The following morning, each liter of M9 media was inoculated with 1 mL of starter culture. M9 minimal media supplemented with appropriate antibiotics, MEM vitamins (Mediatech), and 50 µM $ZnCl_2$. As required, 1 g of $^{15}HN_4Cl$ and 2 g $^{13}C$-glucose (U- $^{13}C_6$, 99%) was included per liter of M9 to produce U-$^{13}C$,$^{15}N$-labeled samples. Cultures were incubated at 22°C with rotary shaking and expression was induced upon reaching an $OD_{600}$ of 0.8 by the addition of 1 mM IPTG. After 24 hr the cells were harvested, and the pellet was stored at −80°C. To produce uniformly-$^2H$,$^{13}C$,$^{15}N$-labeled samples, cells were first conditioned for growth in 100% $D_2O$ medium in the following manner. First, cells were cultured in 3 mL of $D_2O$ LB for 3 to 5 hr at 37°C until flocculent and then used to inoculate 30 mL 100% $D_2O$ M9 media. After 12 to 16 hr of incubation, once the cells reached mid-log growth phase, 15 mL of the culture was used to inoculate 1L of 100% $D_2O$ M9. Large-scale growth continued at 37°C until the $OD_{600}$ reach 0.5, at which point the temperature was adjusted to 22°C. The protocol then proceeded as described for the preparation of double-labeled samples. Appropriate antibiotics were included at all steps.

Amino acid-selective isotopic labeling was performed as previously described (*Peng et al., 2014*). The transaminase deficient strain of *E. coli*, CT19 (a gift from Dr. David Waugh of the US National Cance Institute), was used to reduce $^{15}N$-labeled amino acid scrambling. CT19 cells were transformed with pET16b-Q1-VSD plasmid and grown in 4L of LB media containing 10 mg/L ampicillin, 100 mg/L Kanamycin, and 20 mg/L tetracycline at 37°C. Once the culture reached an $OD_{600}$ of 0.6 the cells were harvested at 2,500 g for 15 min. The pellet was gently resuspended in 1L of M9 media containing 0.2 g of the $^{15}N$-labeled amino acid of interest. Additionally, the media was supplemented with 0.5 g of alanine, phenylalanine, leucine, isoleucine, aspartate, tyrosine, and 0.1 g

tryptophan (excluding the amino acid to be labeled in each case). Culture growth then proceeded as described previously. Four samples were prepared, each with a single $^{15}N$-labeled amino acid, including Val, Leu, Ile, or Phe. Additionally, reverse isotopic labeling of Arg residues was carried out by the addition of excess $^{14}N$-Arg (1 g/L) in $^{15}NH_4Cl$ M9 media prior to induction.

## Protein purification

Cell pellets were resuspended at a ratio of 1 g per 20 mL lysis buffer (75 mM Tris-HCl, 300 mM NaCl, and 0.2 mM EDTA (ethylenediaminetetraacetic acid, pH 7.8) with 5 mM Mg(Ac)$_2$, 0.2 mg/ml PMSF (phenylmethylsulfonyl fluoride), 0.02 mg/ml DNase, 0.02 mg/ml RNase and 0.2 mg/ml lysozyme and tumbled for 30 min. The cells were lysed by probe sonication for 10 min and a cycle time of 10 s on ice at 4°C. Inclusion bodies were then isolated by centrifugation at 20,000 g for 20 min at 4°C. The pellet was resuspended by homogenization and the sonication step was repeated once more. After cell lysis, the pellet was resuspended at a ratio of 1 g pre-lysis pellet weight per 10 mL buffer A (40 mM HEPES, 300 mM NaCl, pH 7.5) containing 0.5% (w/v) dodecylphosphocholine (DPC) (Anatrace, Maumee, OH) and 2 mM TCEP and tumbled overnight at 4°C to solubilize the inclusion bodies. The following morning, insoluble debris was removed by centrifugation at 20,000 g for 20 min. The supernatant was then incubated with 0.2 mL Superflow Ni(II)-NTA (Qiagen, Germantown, MD) per 1 g pre-lysis pellet weight for at least 1 hr at 4°C. After batch binding, the Ni(II)-NTA was then packed into a gravity-flow column and washed with 10 column volumes (CV) of buffer A containing 0.5% (w/v) DPC and 2 mM TCEP. Impurities were eluted by washing with 12 CV of buffer A containing 0.5% DPC, 2 mM TCEP, and 60 mM imidazole (pH 7.5). Detergent exchange was performed by washing the column with 10 CV of buffer A containing 0.2% (w/v) LMPG (lyso-myristoyl-phosphatidylglycerol) and 2 mM TCEP. The KCNQ1-VSD was eluted in buffer A containing 0.2% (w/v) LMPG, 2 mM TCEP, and 500 mM imidazole until A$_{280,}$ as monitored continuously, returned to the baseline level (3–4 CV). Typical Q1-VSD yields were 2–3 mg per liter of growth. The eluent was concentration ten-fold by centrifugation (3700 g, 4°C) in an Amicon Ultra cartridge (10,000 molecular weight cut-off). The sample was then diluted ten-fold in NMR buffer (40 mM MES, 0.5 mM EDTA, 2 mM TCEP, pH 5.5). This process was repeated a total of four times. The KCNQ1-VSD concentration was determined by A$_{280}$ using an extinction coefficient of 34950 M$^{-1}$ cm$^{-1}$. Samples were flash frozen in liquid nitrogen and stored at −80°C.

## Preparation of spin-labeled samples

A cysteine-free KCNQ1-VSD construct was generated with the following amino acid substitutions: C122S, C136A, C180S, and C214A. Combinations of amino acid substitutions were tested to identify the combination that produced only very minimal perturbation of the TROSY-HSQC spectrum relative to the native KCNQ1-VSD. Notably, KCNQ1 cysteine substitutions have been shown to not significantly perturb channel function (Xu et al., 2008). Eight single-cysteine constructs were used for MTSL labeling and PRE measurements: K121C, T144C, T155C, T177C, C180, C214, T224C, and M238C. Each single-cysteine construct was U-$^{15}N$ labeled and purified as previously described (Peng et al., 2014). To the rougly 8 ml of Ni(II)-NTA elution, DTT and EDTA were added to final concentrations of 2.5 mM and 1 mM respectively. The pH was then carefully adjusted from 7.5 to 6.5 by multiple additions of 0.2 mL 1 M HCl. Prior to a 2 hr 25°C incubation with gentle tumbling, the volume was reduced to 0.5 mL by centrifugal ultrafiltration (Amicon Ultra cartridge 10,000 MWCO, 3,700 g, 4°C). Incubation continued overnight after the addition of MTSL (1-oxyl-2,2,5,5-tetramethyl-pyrroline-3-methyl-methanethiosulfonate, Santa Cruz Biotechnology) to 10 mM from a 0.25 M stock in DMSO. Argon gas was used to displace any air within the incubation tube. The following morning, samples were diluted to 10 mL of buffer A (40 mM HEPES, 300 mM NaCl, pH 7.5) and then concentrated 20-fold. After repeating this step, MTSL-labeled protein was batch bound to 1 mL Ni(II)-NTA and incubated for 1 hr. After batch binding, the Ni(II)-NTA was then packed into a gravity-flow column and washed with 16 column volumes (CV) of buffer A containing 0.2% (w/v) LMPG. The sample was eluted and prepared for NMR experiments as previously described (Peng et al., 2014).

## Preparation of aligned Q1-VSD for residual dipolar coupling measurements

A neutral 5% polyacrylamide gel (50:1 acrylamide:bis-acrylamide molar ratio) was polymerized in a cylindrical casing with a 6 mm inner diameter. After two hours, the gel plug was displaced and equilibrated with NMR buffer in three steps. In the first step, the gel was incubated in 50 mL of buffer (40 mM MES, 0.5 mM EDTA, pH 5.5) for six hours. This step was repeated once, and then in the final step the gel was equilibrated against NMR buffer (40 mM MES, 2 mM TCEP, 0.05% LMPG (w/v), 0.5 mM EDTA, 5% $D_2O$, pH 5.5). Subsequently, the gel was then cut to 12 mm in length and transferred to a 1.5 mL cryotube and incubated with ca. 0.6 mL of 0.4 mM $^{15}$N-KCNQ1-VSD for two days. The gel was then stretched into an open-ended 5 mm NMR tube (4.2 mm inner diameter, New Era). The remaining KCNQ1-VSD solution was transferred to a 3 mm NMR tube and used to measure $J_{NH}$ couplings under isotropic conditions.

## NMR data collection and processing

All NMR data were collected at 50°C on Bruker Avance spectrometers at 600 MHz (14.4 T), 800 MHz (18.7 T), or 900 MHz (21.1 T), each equipped with a cryoprobe. NMR data were processed in Topspin 3.2 or qMDD (*Lemak et al., 2011*) and analyzed with NMRFAM-Sparky (*Lee et al., 2015*). NMR samples were composed of 0.2–0.4 mM KCNQ1-VSD, 50 mM MES, 0.5 mM ETDA, 2 mM TCEP, and 50 to 80 mM LMPG. Between 2.5% and 7.5% (v/v) $D_2O$ was added to each sample prior to data acquisition. A shaped tube containing 0.4 mL of 0.4 mM KCNQ1-VSD was used for all triple resonance experiments. Proton chemical shifts were referenced to internal DSS while $^{15}$N and $^{13}$C chemical shifts were referenced indirectly to DSS using absolute frequency ratios. Non-uniform sampling (NUS) was used to increase resolution per unit time of acquisition for triple resonance backbone and side-chain experiments (*Kazimierczuk and Orekhov, 2011*).

## Chemical shift assignments

Backbone $^1$H$^N$, $^{15}$N, $^{13}$C$^a$,$^{13}$C$^b$, and $^{13}$C' chemical shifts were assigned using TROSY versions of three-dimensional (3D) HNCA, HNCO, HN(CO)CA, and HNCACB experiments at 900 MHz on a $^2$H,$^{13}$C,$^{15}$N-KCNQ1-VSD sample (*Loria et al., 1999*). Triple resonance backbone experiments were carried out using NUS with 25% sparse sampling. In addition, an $^{15}$N-edited 3D NOESY-TROSY was recoded with an NOE mixing time ($\tau_{mix}$) of 150 ms at 900 MHz. Two-dimensional (2D) TROSY HSQC spectra were recorded on samples with selectively labeled amino acids to aid in resonance assignments. Backbone amide peaks for 140 out of 147 non-proline residues (95%) were assigned.

Side-chain assignments were based on data from NOESY and amide-correlated TOCSY experiments. 3D TROSY-(H)C(CO)NH-TOCSY and TROSY-H(CCO)NH-TOCSY experiments were carried out using NUS with 50% sparse sampling recorded on a $^{13}$C,$^{15}$N-labeled sample at 600 MHz. Additionally, an H(C)CH-COSY was recorded using NUS with 42% sparse sampling. 2D $^{13}$C-edited HSCQ and 3D $^{13}$C-edited NOESY ($\tau_{mix}$ = 150 ms) experiments were recorded on a uniformly $^{13}$C,$^{15}$N-labeled sample in 99% $D_2O$ (v/v) at 900 MHz. A 3D $^{15}$N-edited NOESY ($\tau_{mix}$ = 120 ms) at 900 MHz was collected using an $^{15}$N-labeled sample. Additionally, methyl optimized 2D $^{13}$C-edited HSCQ and 3D $^{13}$C-edited NOESY ($\tau_{mix}$ = 200 ms) experiments were recorded on a $^{13}$C,$^{15}$N-labeled sample in deuterated LMPG (FBReagents) and 99% $D_2O$ (v/v) at 900 MHz. Methyl groups for 58 out of 68 (85%) residues were assigned (Ala 10/10, Thr 6/8, Ile 9/11, Val 18/22, Leu 15/17).

## Paramagnetic relaxation enhancements (PREs)

PRE measurements were carried out as described previously (*Deatherage et al., 2017*). Briefly, two TROSY HSQC spectra were acquired with matched parameters and processed identically on each spin-labeled sample at 900 MHz. The first spectrum was collected under paramagnetic conditions and then reduced by the addition of pH-matched ascorbic acid to 20 mM. Importantly, the change in sample volume after addition of ascorbic acid was under 1%. The second spectrum was then collected under diamagnetic conditions. The intensity ratios and the diamagnetic sample linewidths were used to determine distances between the backbone amide proton and site of the spin label (*Battiste and Wagner, 2000*).

## Residual dipolar couplings (RDCs)

Backbone $^1$H-$^{15}$N RDC data was acquired by measuring an HSQC and TROSY spectrum for the aligned and isotropic KCNQ1-VSD samples. The $^1$H couplings were obtained from doubling the resonance frequency difference between the HSQC and TROSY peaks in the $^{15}$N dimension. In the isotropic and aligned samples this frequency difference corresponds to $J_{NH}$ and $J_{NH}+D_{NH}$ respectively. The initial estimates of the axial (Da) and rhombic (R) components of the alignment tensor were derived from the largest observed $D_{NH}$-value and the program calcTensor (distributed with XPLOR-NIH) using the KCNQ1-VSD ensemble determined in the absence of RDC restraints (*Schwieters et al., 2006*).

## Structure calculations

Structure calculations were performed using XPLOR-NIH via simulated annealing protocols in three steps (*Schwieters et al., 2006*), as summarized in this paragraph. In the first step, starting with an extended polypeptide the secondary structure was defined using local NOE-derived distance and backbone torsion restraints. Hydrogen bond distance and geometry restraints were then incorporated based on two criteria (i) observed helices in the initial ensemble, and (ii) chemical shift index analysis (*Wishart and Sykes, 1994*). In the second step, long-range NOE- and PRE- derived restraints were added to define the tertiary contacts of the VSD. First local NOEs were implemented, contributing to identification of well-defined secondary structural elements. Then high confidence long-range NOEs were incorporated, resulting in a loosely defined ensemble. Additional long-range NOEs were added through an iterative process until a precise ensemble was achieved. As described at the end of this section of the Methods, care was taken to verify the structural outcome was not unduly overdependent on any subset of long-range NOE data. In the third and final step, the top 10% of the structural ensemble were refined with RDC data. Details for all three steps are as follows.

Backbone torsion angle restraints were determined using $^{15}$N, $^{13}$C′, $^{13}$C$^\alpha$, and $^{13}$C$^\beta$, chemical shifts using the program TALOS-N (*Shen and Bax, 2013*). Only dihedral angle restraints classified as 'strong' with a confidence score of 7 or higher were used with an error set to 20°. Resonances in transmembrane helices S1, S2, and S3, exhibited significantly reduced peak amplitudes in 2D spectra as compared to the remainder of the protein. Based on this observation, NOE cross-peaks were divided into two groups and calibrated separately. NOESY spectra cross-peaks were manually assigned (see 'Chemical Shifts Assignment' section) and classified based on intensity distribution into one of four groups: strong, medium, weak, and very weak. These classifications correspond to distance restraints of 1.8–2.8, 1.8–3.5, 1.8–4.5, and 1.8–5.5 angstroms. PRE restraints were implemented as the distance between the CB of the spin-labeled residue to the backbone amide hydrogen. Each restraint was categorized into one of three bins based on the intensity ratio between the paramagnetic and diamagnetic ($I_{para}/I_{dia}$) spectra. For an $I_{para}/I_{dia}$ ratio of less than 0.2 the distance was restrained to between 2 and 18 Å. When the $I_{para}/I_{dia}$ ratio was between 0.2 and 0.8 an explicit distance was calculated and given an uncertainty of $\pm6$ Å[56]. Lastly, an $I_{para}/I_{dia}$ ratio great than 0.8 was restrained to between 19 to 100 Å. Only isolated NMR resonances were selected to be incorporated as distance restraints in structure calculations. PREs were observed for S0 in experiments when the spin label was located on either the intra- or extra-cellular face of the VSD, suggesting the S0 helix undergoes motions making the inclusion of PRE-derived distance restraints to this structural element inappropriate. For this reason, no PRE restraints were used for residues within S0.

Structure calculations were conducted in a similar manner to previous work, using standard XPLOR-NIH protocols, where calculations are carried out in four stages: high temperature molecular dynamics, simulated annealing, torsion angle, and Cartesian minimization (*Deatherage et al., 2017*). In the first stage, the temperature was set to 3,500 K for 20 ps with the following force constants: $k_{bond\ angle}$=0.4 kcal mol$^{-1}$ deg$^{-1}$; $k_{improper}$ = 0.4 kcal mol$^{-1}$ deg$^{-2}$; $k_{backbone\ dihedrals}$=5 kcal mol$^{-1}$ rad$^{-1}$; $k_{NOE}$ = 20 kcal mol$^{-1}$ Å$^{-1}$; $k_{PRE}$ = 2 kcal mol$^{-1}$ Å$^{-1}$. Prior to simulated annealing, force constants were increased over two ps: $k_{atom\ radii}$ = 0.4-fold to 0.8 fold; $k_{van\ der\ waals}$=0.004–4 kcal mol$^{-1}$ Å$^{-2}$; $k_{bond\ angle}$=0.4–1.0 kcal mol$^{-1}$ deg$^{-1}$; $k_{improper}$ = 0.4–1.0 kcal mol$^{-1}$ deg$^{-2}$. During simulated annealing, the temperature was reduced to 100 K in 25 K steps while force constants were increased, $k_{NOE}$ = 20–30 kcal mol$^{-1}$ Å$^{-1}$, $k_{PRE}$ = 2–3 kcal mol$^{-1}$ Å$^{-1}$, and $k_{backbone\ dihedrals}$ was set to 200 kcal mol$^{-1}$ rad$^{-1}$. Distance restraints were enforced by a flat well harmonic and H-bond potentials were included. A total of 150 structures were calculated and the top 15 lowest energy structures

were refined with RDC data. In refinement, the bath temperature was set to 3000 K for 10 ps and then cooled to 25 K in 12.5 steps. RDC restraints were set to 0.05 kcal mol$^{-1}$ Hz$^{-2}$ for the high temperature phase and then ramped to 0.8 kcal mol$^{-1}$ Hz$^{-2}$ during simulated annealing. A total of 150 structures were calculated. The RDC refined ensemble is shown in *Figure 2—figure supplement 1A* and its structure statistics are presented in *Table 1*.

XPLOR-NIH calculations were followed by data-restrained molecular dynamics (rMD) refinement in an explicit lipid bilayer using the AMBER16 force field. This step allows the micellar structure to be adjusted in a simulated bilayer to adjust for any micellar distortions, while still enforcing the NMR data restraints. For this, 10 structures were selected from the RDC-refined XPLOR-NIH ensemble based on r.m.s.d. to the mean coordinates and consistency of S0 orientation with previous experimental results (*Sun and MacKinnon, 2017*). These 10 structures then were solvated in an explicit DMPC bilayer using the CHARMM-GUI server (*Wu et al., 2014*). Notably, a DMPG bilayer was also tested as a rMD environment and no significant differences in restraint violations as compared to DMPC were observed. Simulations used the Lipid17 AMBER lipid force field and the ff14SB force field (*Dickson et al., 2014*). Using GPU-accelerated AMBER16 (*Case et al., 2005*), restrained minimization of each structure was performed stepwise. Over 30,000 steps first the lipids and then the aqueous solvent was minimized with the protein atoms restrained to initial positions. Then over 20,000 steps the protein and subsequently the entire system was minimized with NMR restraints. The force constants for distance and angle restraints were set to 10 kCal/mol/Å (*Bezanilla et al., 1994*) and 20 kCal/mol/rad (*Bezanilla et al., 1994*) respectively. The system was then heated to 323K in six steps. In the first step, the system was heated to 50K and the protein backbone, sidechains, lipid head groups, lipid tails, and ions were each restrained with force constants of 10, 5, 2.5, 2.5 and 10 kCal/mol/Å (*Bezanilla et al., 1994*), respectively. In the subsequent five steps, the system was heated from 50 K to 323 K iteratively. In each repetition the force constants were reduced until the final round where only the protein backbone was restrained with a force constant of 0.1 kCal/mol/Å (*Bezanilla et al., 1994*). The system was then equilibrated for five ns with protein atoms constrained to starting position. After equilibration, all atoms were released, and NMR restraints were ramped to 100% wt over 20 ps. Each trajectory ran with NMR restraints for a total 100 ns using constant pressure periodic boundary conditions and anisotropic pressure scaling. Each trajectory was then extended at least another 190 ns without NMR restraints, with the results then being tested to verify they remained consistent with the NMR restraints. The unrestrained trajectory seeded with the lowest energy rMD structure shifted only 3 Å for all residues and 2 Å for transmembrane helices (*Figure 2—figure supplement 1D,E*), leading to the final ensemble, (PDB ID: 6MIE). The final ensemble is composed of the centroid of the most populated cluster for each 10 ns block of time over the last 100 ns of the trajectory and is shown in *Figure 2—figure supplement 1B*. Clustering was preformed using the dbscan algorithm in CPPTRAJ (*Case et al., 2005*). Each member of this final ensemble was then scored with the NMR restraints and found to be consistent with experimental data (*Table 1*). A comparision of the XplorNIH and MD-refined ensemble is shown in *Figure 2—figure supplement 1*.

To verify that the ensemble was not dependent on small subset of the NOE restraints, structure calculations were repeated where a random fraction of long-range NOE restraints were excluded (10%). This was repeated a total of 10 times, where in each run a different fraction of data was withheld, such that all long-range NOE-derived restraints were excluded at least once. In all of these calculations, the same overall fold and ion pairings of gating residues were observed as in the final reported structure. The primary difference between the 10 runs was the precision to which the ensemble was determined, which demonstrates that the final structures (represented by PDB 6MIE, see also *Figure 2*) are not dependent on any specific subset of long-range NOE-derived restraints.

## Oocyte expression

Stage V or VI oocytes were obtained from *Xenopus laevis* by laparotomy. All procedures were performed in accordance with the protocol approved by the Washington University Animal Studies Committee (Protocol # 20190030). Oocytes were digested by collagenase (0.5 mg/ml, Sigma Aldrich, St Louis, MO) and injected with channel cRNAs (Drummond Nanoject, Broomall). Each oocyte was injected with cRNAs (9.2 ng) of WT or mutant KCNQ1, with or without KCNE cRNAs (2.3 ng). Injected cells were incubated in ND96 solution (in mM): 96 NaCl, 2 KCl, 1.8 CaCl$_2$, 1 MgCl$_2$, 5 HEPES, 2.5 CH$_3$COCO$_2$Na, 1:100 Pen-Strep, pH 7.6) at 18°C for at least 2 days before recording.

## Two-electrode voltage clamp (TEVC) and voltage-clamp fluorometry (VCF)

Microelectrodes (Sutter Instrument, Item #: B150-117-10) were made with a puller (Sutter Instrument, P-97), and the resistances were 0.5–3 MΩ when filled with 3 M KCl solution. Ionic currents were recorded by TEVC in ND96 bath solutions. Whole-oocyte currents were recorded using a CA-1B amplifier (Dagan, Minneapolis, MN) with Patchmaster (HEKA) software. The currents were sampled at 1 kHz and low-pass-filtered at 2 kHz. All recordings were carried out at room temperature (21–23°C). For experiments comparing the current amplitude of the KCNQ1 channel with and without KCNE1 co-expression, steps were taken to control oocyte channel expression that can confound current amplitude comparison. RNAs encoding for each mutant KCNQ1 channel were injected the same day, with and without KCNE1 RNAs co-injection. The cells injected with the same mutant KCNQ1 RNA were later recorded during the same day after channel expression. This controls for channel expression for each mutant with and without KCNE1 co-expression and allows for current amplitude comparison within each mutant. For XE991 and chromanol 293B experiments, the cells were held at −20 mV holding potential and pulsed to +40 mV (4 s) and −40 mV (2 s) every 20 s. Each cell was first recorded under control ND96 solution until steady state. Stock XE991 (10 mM) and Chromanol 293B (100 mM) were added after ionic current in control and XE991 solutions reached steady state, respectively. Stock drugs were added to achieve final dilution of 5 μM XE991 and 150 μM chromanol 293B. All cRNA amounts were doubled for VCF experiments to achieve higher surface expression level. Oocytes were incubated for 30 min on ice in 10 μM Alexa 488 C5-maleimide (Molecular Probes, Eugene, OR) in high $K^+$ solution in mM (98 KCl, 1.8 CaCl$_2$, 5 HEPES, pH 7.6) for labeling. Cells were washed three times with ND96 solution to remove the labeling solution, and recordings were performed in ND96 solution on the CA-1B amplifier setup. Excitation and emission lights were filtered by a FITC filter cube (Leica, Germany, for Alexa 488) and the fluorescence signals were collected by a Pin20A photodiode (OSI Optoelectronics). The signals were then amplified by an EPC10 (HEKA, analog filtered at 200 Hz, sampled at 1 kHz) patch clamp amplifier and controlled by the CA-1B amplifier to ensure fluorescence signals were recorded simultaneously with currents. All other chemicals were from Sigma Aldrich.

## Electrophysiology data analysis

Data were analyzed with IGOR (Wavemetrics, Lake Oswego, OR), Clampfit (Axon Instruments, Inc, Sunnyvale, CA), Sigmaplot (SPSS, Inc, San Jose, CA), and custom MATLAB (MathWorks, MA) software. The instantaneous tail currents following test pulses were normalized to the maximal current to calculate the conductance-voltage (G-V) relationship. Because of photo-bleaching, fluorescence signals were baseline subtracted by fitting and extrapolating the first 2 s signals at the −80 mV holding potential. ΔF/F was calculated after baseline subtraction. Fluorescence-voltage (F-V) relationships were derived by normalizing the ΔF/F value at the end of each four-seconds test pulse to the maximal value. F-V and G-V curves were fitted with either one or the sum of two Boltzmann equations in the form $1/(1+\exp(-z*F*(V-V_{1/2})/RT))$ where $z$ is the equivalent valence of the transition, $V_{1/2}$ is the voltage at which the transition is half maximal, $R$ is the gas constant, $T$ is the absolute temperature, $F$ is the Faraday constant, and $V$ is the voltage. Current inhibition from XE991 was calculated by using the steady-state current amplitude at the end of the four-seconds test pulse in control ($I_{control}$) and drug ($I_{XE991}$ or $I_{chromanol}$) solutions. The fraction of XE991 inhibition was calculated by first subtracting $I_{chromanol}$ from $I_{control}$ and $I_{XE991}$ to account for endogenous current contamination. XE991 inhibition fraction was then calculated utilizing the ratio of the chromanol-subtracted current with the following equation:

$$f_{XE991} = 1 - \frac{I_{XE991} - I_{chromanol}}{I_{control} - I_{chromanol}} = \frac{I_{control} - I_{XE991}}{I_{control} - I_{chromanol}} \tag{1}$$

## Acknowledgements

We thank Dr. Ben Mueller for his assistance with the software used in the structural studies, Dr. Kirill Oxenoid for advice on NMR experiments, and Prof. Markus Voehler for technical assistance with NMR experiments. This work was supported by NIH R01 HL122010 (to CRS, ALG, and JM), by R01 NS092570 and R01 HL126774 to JC, by AHA 18POST34030203 to PH and by NIH F32 GM117770 (to KCT).

## Additional information

### Competing interests
Jingyi Shi, Jianmin Cui: is the cofounder of a startup company VivoCor LLC, which is targeting IKs for the treatment of cardiac arrhythmia. The other authors declare that no competing interests exist.

### Funding

| Funder | Grant reference number | Author |
| --- | --- | --- |
| National Institutes of Health | R01 HL122010 | Alfred L George<br>Jens Meiler<br>Charles R Sanders |
| National Institutes of Health | R01 NS092570 | Jianmin Cui |
| National Institutes of Health | R01 HL126774 | Jianmin Cui |
| National Institutes of Health | F32 GM117770 | Keenan C Taylor |
| American Heart Association | 18POST34030203 | Panpan Hou |

The funders had no role in study design, data collection and interpretation, or the decision to submit the work for publication.

### Author contributions
Keenan C Taylor, Po Wei Kang, Panpan Hou, Data curation, Formal analysis, Validation, Investigation, Visualization, Methodology, Writing - original draft, Writing - review and editing; Nien-Du Yang, Data curation, Formal analysis, Validation, Investigation, Writing - original draft; Georg Kuenze, Software, Methodology, Writing - original draft, Writing - review and editing; Jarrod A Smith, Software, Methodology; Jingyi Shi, Data curation, Formal analysis, Investigation, Writing - original draft; Hui Huang, Investigation, Methodology, Writing - original draft, Writing - review and editing; Kelli McFarland White, Formal analysis, Investigation; Dungeng Peng, Investigation, Methodology, Writing - original draft; Alfred L George, Funding acquisition, Writing - original draft, Writing - review and editing; Jens Meiler, Supervision, Funding acquisition, Writing - original draft, Writing - review and editing; Robert L McFeeters, Formal analysis, Methodology, Writing - original draft; Jianmin Cui, Charles R Sanders, Conceptualization, Data curation, Supervision, Funding acquisition, Methodology, Writing - original draft, Project administration, Writing - review and editing

### Author ORCIDs
Po Wei Kang  https://orcid.org/0000-0002-5933-545X
Panpan Hou  https://orcid.org/0000-0001-7694-2262
Nien-Du Yang  http://orcid.org/0000-0002-5261-7382
Jianmin Cui  https://orcid.org/0000-0002-9198-6332
Charles R Sanders  https://orcid.org/0000-0003-2046-2862

### Ethics
Animal experimentation: Oocytes from *Xenopus laevis* (frogs) were employed in this work (at Washington University) and the frogs were cared for in accordance with the protocol approved by the Washington University Animal Studies Committee (Protocol # 20190030).

### Decision letter and Author response
Decision letter https://doi.org/10.7554/eLife.53901.sa1
Author response https://doi.org/10.7554/eLife.53901.sa2

## Additional files

### Supplementary files
• Transparent reporting form

### Data availability

The structures determined in this work have been deposited into the Protein Databank (PDB ID 6MIE). NMR data assignments and structural restraints have been deposited in the BioMagResBank (BMRB ID 30517). All electrophysiology and voltage-clamp fluorometry data generated or analysed during this study are included in the manuscript, supporting files, and source data file. All data needed to evaluate the conclusions in the paper are present in the paper and/or in the figure supplements and source data files. Correspondence and request for materials should be addressed to JC (jcui@wustl.edu) or CRS (chuck.sanders@vanderbilt.edu).

The following datasets were generated:

| Author(s) | Year | Dataset title | Dataset URL | Database and Identifier |
|---|---|---|---|---|
| Taylor KC, Kuenze G, Smith JA, Meiler J, McFeeters RL, Sanders CR | 2018 | NMR structure of the KCNQ1 voltage-sensing domain | http://www.rcsb.org/structure/6MIE | RCSB Protein Data Bank, 6MIE |
| Taylor K, Kuenze G, Smith J, Meiler J, McFeeters R, Sanders CR | 2018 | Solution NMR structure of the KCNQ1 voltage-sensing domain | http://www.bmrb.wisc.edu/data_library/summary/index.php?bmrbId=30517 | Biological Magnetic Resonance Data Bank, 30517 |

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
