## [Decision Letter]

**Acceptance summary:**

This paper provides a high-resolution structure of a voltage sensor domain from a human potassium channel, KCNQ1. The structure reveals an "intermediate" state, which has been studied functionally for 25 years but not seen at high resolution until now. Electrophysiological analyses provide key validation of the new structure, and voltage clamp fluorometry demonstrates the physiological relevance of the intermediate state for KCNQ1 channels.

**Decision letter after peer review:**

Thank you for submitting your article "Structure and physiological function of the KCNQ1 channel voltage sensor intermediate state" for consideration by *eLife*. Your article has been reviewed by four peer reviewers, including Merritt Maduke as the Reviewing Editor and Reviewer #1, and the evaluation has been overseen by Richard Aldrich as the Senior Editor. The following individual involved in review of your submission has agreed to reveal their identity: Lucie Delemotte (Reviewer #3).

The reviewers have discussed the reviews with one another and the Reviewing Editor has drafted this decision to help you prepare a revised submission.

Summary:

This manuscript presents a structure of the Kv7.1 voltage sensor domain in LMPG micelles, solved by solution-state NMR. Based on the positioning of helices S4 and S2, the structure appears to be that of an "intermediate" state, which has not been observed in any previous structures. To test this prediction, the authors use a "double charge reversal mutagenesis strategy" and measure effects on channel gating and inhibition electrophysiologically. The data nicely show that mutants have the predicted effects on either the intermediate or activated functional states. The authors additionally use voltage clamp fluorimetry to investigate how accessory subunits KCNE3 and KCNE1 affect structure and function of KCNQ1 channels, demonstrating that KCNQ1/KCNE3 channels open to the intermediate state, while KCNQ1/KCNE1 channels open to the fully activated state.

Essential revisions:

The essential revisions concern the NMR structure determination and the presentation of the electrophysiological data. For the NMR (points 1-4), additional information is required for evaluation of the structure. For the electrophysiology (points 5-8), clarifications are required, and an additional experiment is suggested but not required.

NMR:

Negatively charged LMPG detergent is itself a cause of great concern. This detergent, although generally yields high quality HSQC spectra for many α-helical membrane proteins, does not have a good reputation for preserving the native or functionally relevant state of membrane proteins. For this reason, the use of LMPG draws much greater scrutiny in the NMR field. On the other hand, it is also the use of LMPG, as the authors suggested, that somehow "stabilized" the elusive intermediate state of the VSD, and they have functional mutagenesis data to support this claim. Hence, rather than refuting LMPG by default, we choose to focus on whether the reported structure in LMPG is truly determined by the unambiguous NMR data.

1) How reliable are the assignments of the long-range NOEs? The methyl spectrum obviously does not have great CS dispersion. It would be helpful to explain in the Materials and methods how long-range NOEs were assigned. The authors should show representative NOE stripes that reflect the quality of the NOE data.

2) In particular, the critical long-range interactions, R2 – E1, Q3 – F0, and R4 – E2, appear to be extremely well defined in Figure 4A. Are the sidechains defined by experimental NOE restraints or by MD simulation? If the former is true, please show NOE stripes for these interactions, as it would strengthen the structural conclusion of the paper. If the latter is true, the authors need to compare the structures before and after MD simulation so that we can have a rough idea of how much of the structure was generated by NOE data and how much was due to computer simulation.

3) In the 200 ns of MD simulation in the absence of NMR restraints, how much, if any, did the structure drift? It would be good to provide this information in the SI.

4) RDC refinement is quite tricky with only NH couplings, because XPLOR-NIH can often tweak the NH bonds of a helix to fit the NH RDCs without having to change the helix orientation. To evaluate whether RDC refinement actually did anything, the authors should compare the structures before and after RDC refinement.

Electrophysiology:

5) The rationale for why the double mutants are predicted to stabilize the structures should be spelled out in order to make the manuscript accessible to readers who don't work on voltage sensors.

6) In general, the description of the electrophysiological experiments is dense and difficult to follow. The schematics in Figure 5 should be more thoroughly explained, and the meaning of the green and red arrows should be included in the legend. In Figure 6D-F, the offsets in the traces make it difficult to understand. It would be better to have a larger figure with data traces for the different conditions shown without offsets, and the legend should contain a more complete explanation of how the experiment was done (e.g. when was XE991 applied) and how inhibition was calculated.

7) Why does KCNQ1/KCNE1 seem to be so inhibited by XE991 in Figure 7D-E, if XE991 is specifically inhibiting the intermediate state?

8) As written, the last part involving VCF is quite disconnected from the rest of the paper. One would expect to see VCF on the mutants probed in the patch clamp assays with E1, chromanol/XE991, to fully verify that the state has been trapped (that there is no change in fluorescence with voltage). Ideally, we would like to see such an experiment. However, it is suggested and not required, as the data from Figures 5 and 6 together are already quite compelling.

We do require that you provide a more detailed explanation of how the KCNE3/KCNE1 experiments connect to the story. Additional background information will make the findings accessible to a broader audience.

9) Figures should use colors that are color-blind friendly. Information on color blind-accessible design is available in "Color Universal Design (CUD)" by Masataka Okabe and Kei Ito (http://jfly.iam.u-tokyo.ac.jp/color/) and a palette of unambiguous colors is available at http://jfly.iam.u-tokyo.ac.jp/color/#pallet. In addition, please note that Adobe Photoshop enables users to proof images to ensure accessibility to individuals with color vision impairment.

---

## [Author Response]

Essential revisions:The essential revisions concern the NMR structure determination and the presentation of the electrophysiological data. For the NMR (points 1-4), additional information is required for evaluation of the structure. For the electrophysiology (points 5-8), clarifications are required, and an additional experiment is suggested but not required.NMR:Negatively charged LMPG detergent is itself a cause of great concern. This detergent, although generally yields high quality HSQC spectra for many α-helical membrane proteins, does not have a good reputation for preserving the native or functionally relevant state of membrane proteins. For this reason, the use of LMPG draws much greater scrutiny in the NMR field. On the other hand, it is also the use of LMPG, as the authors suggested, that somehow "stabilized" the elusive intermediate state of the VSD, and they have functional mutagenesis data to support this claim. Hence, rather than refuting LMPG by default, we choose to focus on whether the reported structure in LMPG is truly determined by the unambiguous NMR data.

We appreciate this perspective. I note that when we started this project 5 years ago that I advised the postdoc first author (KCT) that it was entirely possible that a structure determined in micelles (of any kind) would lead to a non‐physiological structure and that we would not know one way or the other till near the end of structure determination, which was projected to take several years. It is much to the credit of the spirit of scientific adventure that he nevertheless chose to undertake this project. We are happy that the structure does indeed appear to be relevant, as demonstrated by its consistency with a variety of previous literature biophysical and structure‐function observations and by the extensive additional experimental validation studies carried out for this manuscript by our Cui lab co‐authors.

In both the Results section and in the Materials and methods we have added additional text to be more clear about the imperative to factor possible micelle‐induced structure perturbations into our approach to structure determination and to be more clear about exactly how this was carried out.

1) How reliable are the assignments of the long-range NOEs? The methyl spectrum obviously does not have great CS dispersion. It would be helpful to explain in the Materials and methods how long-range NOEs were assigned. The authors should show representative NOE stripes that reflect the quality of the NOE data.

Representative NOE strips displaying the quality of data are now shown in Figure 1—figure supplement 3. In the Materials and methods we have included a description of how long‐range NOEs were assigned (subsection “NMR data collection and processing”, subsection “Structure calculations”, first paragraph). We note that care was taken to verify that the ensemble was not overly dependent on a subset of NOE data (it was not, see the last paragraph of the aforementioned subsection”). This is a standard approach in NMR structure determination to minimize the chance that one or a few incorrectly‐assigned restraints could significantly alter the outcome of structure determination.

2) In particular, the critical long-range interactions, R2 – E1, Q3 – F0, and R4 – E2, appear to be extremely well defined in Figure 4A. Are the sidechains defined by experimental NOE restraints or by MD simulation? If the former is true, please show NOE stripes for these interactions, as it would strengthen the structural conclusion of the paper. If the latter is true, the authors need to compare the structures before and after MD simulation so that we can have a rough idea of how much of the structure was generated by NOE data and how much was due to computer simulation.3) In the 200 ns of MD simulation in the absence of NMR restraints, how much, if any, did the structure drift? It would be good to provide this information in the SI.

Figure 2—figure supplements 1A and B show the structure ensemble after the XPLOR‐NIH calculations (panel A) and following the restrained and unrestrained MD trajectories (panel B). Also Figure 2—figure supplement 1D shows the backbone R.M.S.D. to starting coordinates for the final unrestrained trajectory. Over the 190 ns of unrestrained MD simulation the average drift was approximately 2 and 3 angstroms for the transmembrane and all residues, respectively.

The reviewers raise a good point. The critical sidechain interactions, R2‐E1, Q3‐F0, and R4‐E2 were not directly defined by the NMR data because no long‐range NOEs to those side chains could be confidently assigned. Rather the NMR structural data set defines the fold of the protein fold (and some side chain conformations) which, along with the force field used during the molecular dynamics determines the conformations of these of key residues. This is now pointed out in the text (subsection “NMR structure of the KCNQ1 voltage sensor domain”, fourth paragraph). It is significant, however, to note that Author response image 1 shows the conformations of these key sidechain residues following Xplor‐NIH NMR structure determination and prior to any MD refinement. Even at this stage of the structural calculations the ensemble clearly shows E1 is aligned with R2 (as expected for the intermediate state VSD), and not R4 (as seen in the activated state cryo‐EM structure). Moreover, the other highlighted residues are also aligned pairwise with the expected partner as expected for the intermediate state. We also emphasize that Table 1 provides the structural statistics both before and after MD refinement, verifying that final post‐MD ensemble continues to satisfy the NMR data. Finally, please see the new Figure 2—figure supplement 1, which provides additional comparisons of the ensemble before and after MD refinement.

**Author response image 1. respfig1:** XplorNIH NMR ensemble prior to MD refinement displaying the orientation of the key sidechain residues. The S0 and S1 helices are not shown to improve clarity.

4) RDC refinement is quite tricky with only NH couplings, because XPLOR-NIH can often tweak the NH bonds of a helix to fit the NH RDCs without having to change the helix orientation. To evaluate whether RDC refinement actually did anything, the authors should compare the structures before and after RDC refinement.Electrophysiology:5) The rationale for why the double mutants are predicted to stabilize the structures should be spelled out in order to make the manuscript accessible to readers who don't work on voltage sensors.

We made two changes to more clearly present the double mutant experimental strategy. First, we added a paragraph within the main text to explain why the double charge reversal mutations stabilize distinct VSD conformations (subsection “Functional validation of distinct KCNQ1 voltage sensor domain structures”, first paragraph). Second, we modified Figure 5A‐C to add in cartoon schematics to illustrate the rationale of the double mutant experiments (please see response to comment 6 as well).

The main impact of the RDC refinement is the reduction in RMSD of the S0 helix. Please see Author response image 2 for a comparison of XplorNIH ensemble before and after RDC refinement. Table 1 has been updated to include structural statistics for the Xplor‐NIH ensemble after RDC refinement, which shows the distribution of phi psi is favorable.

**Author response image 2. respfig2:** XplorNIH ensemble before (left) and after (middle) RDC refinement. The RDC refined ensemble is colored gray and the NOE/PRE derived ensemble is colored teal.

6) In general, the description of the electrophysiological experiments is dense and difficult to follow. The schematics in Figure 5 should be more thoroughly explained, and the meaning of the green and red arrows should be included in the legend. In Figure 6D-F, the offsets in the traces make it difficult to understand. It would be better to have a larger figure with data traces for the different conditions shown without offsets, and the legend should contain a more complete explanation of how the experiment was done (e.g. when was XE991 applied) and how inhibition was calculated.

These are excellent suggestions. We have taken steps to revamp both figures to increase clarity. Changes to Figures 5 and 6 are detailed below.

Figure 5: Related to comment 5, we added cartoon schematics in Figure 5 to illustrate the rationale for the double mutation experiments and the predicted effect on VSD conformation. We hope these schematics help make the manuscript more accessible.

Figures 5 and 6: We replaced the kinetic models in Figures 5‐6 detailing the effects of KCNE1 and XE991 on distinct KCNQ1 open states with tables. The tables are more readable and unambiguously indicate the expected effect of KCNE1 and XE991 on the pore. We also cite the appropriate studies in the figure legend to support the predictions in the table.

Figure 6: To make clear how the XE991 pharmacology experiments were carried out, we expanded Figure 6 by adding exemplar diary plots of the drug experiments. We hope these diary plots give the readers a better sense of our experimental procedures.

Figure 6: We separated the current traces under different conditions shown in the original Figure 6 for better visualization. We also clearly labeled the current traces within the context of the diary plots with arrows.

Comment 6 also indicates that the XE991 current overlays contained offsets that made it difficult to understand. We agree with the reviewers. However, we did not offset the traces from 0 µA. We performed the drug studies with holding potential at ‐20 mV instead of the typical ‐80 mV, the perceived current “offsets” thus reflect K^+^ currents for the constitutively opened mutants at ‐20 mV. We added our voltage protocol in the main text, figure, and legend to help resolve this confusion.

Figure 6: The % XE inhibition was calculated by comparing the steady‐state current at the end of the +40 mV test pulse after chromanol current subtraction. To make clear how XE991 inhibition was calculated, we added the chromanol‐subtracted currents for control and XE991 conditions in Figure 6 across all mutants. We also added texts in the Materials and methods (subsection “Electrophysiology data analysis”) to explain how the inhibition was calculated.

In addition to the figure changes, we also expanded the main text to describe details underlying our experimental procedures and interpretations (subsection “Functional validation of distinct KCNQ1 voltage sensor domain structures”). We devoted particular attention to detailing the logic underlying our mutagenesis strategy (related to comment 5) and the XE991 pharmacology experimental procedure and inhibition calculation.

7) Why does KCNQ1/KCNE1 seem to be so inhibited by XE991 in Figure 7D-E, if XE991 is specifically inhibiting the intermediate state?8) As written, the last part involving VCF is quite disconnected from the rest of the paper. One would expect to see VCF on the mutants probed in the patch clamp assays with E1, chromanol/XE991, to fully verify that the state has been trapped (that there is no change in fluorescence with voltage). Ideally, we would like to see such an experiment. However, it is suggested and not required, as the data from Figures 5 and 6 together are already quite compelling.We do require that you provide a more detailed explanation of how the KCNE3/KCNE1 experiments connect to the story. Additional background information will make the findings accessible to a broader audience.

This is a good question. Our previous studies demonstrated that although the AO state is resistant to XE991 inhibition, KCNE1 re‐sensitizes the AO state to permit some XE991 inhibition. However, significant differences in XE991 blocking kinetics, the time‐dependent inhibition during each pulse of current activation, be detected between KCNQ1 and KCNQ1/KCNE1 (Zaydman et al., 2014). This difference is demonstrated in Figure 7E (previously Figure 7D) in which the KCNQ1/KCNE1 current inhibition only reaches the level of KCNQ1 block at 1 second, but shows much less inhibition at 300 ms (300 ms is action potential duration in human heart cells, see Author response image 3, indicating that KCNQ1/KCNE1 primarily conducts at the AO state. We added clarification for this in the main text (subsection “Physiological role of the intermediate state of the KCNQ1 voltage sensor”, last paragraph).

**Author response image 3. respfig3:** 5 µM XE991 inhibition of KCNQ1, KCNQ1+KCNE1, and KCNQ1+KCNE3 channels at 300 ms activation duration. % inhibition was calculated with 1 – (I_XE991_ / I_Control_) at 300 ms after the test pulse started.

We have also added paragraphs in the main text to better connect the KCNE1/KCNE3 experiments/figure to the overarching story (subsection “Physiological role of the intermediate state of the KCNQ1 voltage sensor”). We more clearly explain the rationale for our experiments to show that we examined whether the KCNQ1 intermediate VSD state and the associated IO state is physiologically relevant. We pointed out that KCNE1 suppression of the IO state implies that the AO state, but not the IO state, plays a role in cardiac physiology. We thus looked to other tissues such as epithelial cells to show that KCNE3 makes KCNQ1 constitutively open to the IO state at the physiological voltage ranges. This indicates that the IO state plays a role in epithelial physiology and that both the IO and AO states are physiologically important.

9) Figures should use colors that are color-blind friendly. Information on color blind-accessible design is available in "Color Universal Design (CUD)" by Masataka Okabe and Kei Ito (http://jfly.iam.u-tokyo.ac.jp/color/) and a palette of unambiguous colors is available at http://jfly.iam.u-tokyo.ac.jp/color/#pallet. In addition, please note that Adobe Photoshop enables users to proof images to ensure accessibility to individuals with color vision impairment.

Done for all figures.

We appreciate the reviewers’ comments. As suggested, we performed new VCF experiments on E1R/R2E and F0R/H5E (see Author response image 4). From the results, E1R/R2E shows constitutive currents but little VSD movement, supporting that the VSD is trapped; while F0R/H5E still shows VSD movement, and a left‐shifted G‐V relation (see tail currents), which is consistent with Figure 5—figure supplement 1. The F0R/H5E result can be a good positive control for E1R/R2E. On the other hand, the unchanged fluorescence for E1R/R2E is a negative result for which we have no other independent validation regarding whether it is real or a failed experiment. We therefore decide not to include these results in the manuscript. However, we are open to reviewers’ suggestions on this issue.

**Author response image 4. respfig4:** VCF results of E1R/R2E (A) and R0R/H5E (B).